# Co-occurrence networks reveal more complexity than community composition in resistance and resilience of microbial communities

Cheng Gao [1,2,10 ✉], Ling Xu [2,3,10], Liliam Montoya [2], Mary Madera[2], Joy Hollingsworth[4], Liang Chen[5], Elizabeth Purdom [6], Vasanth Singan [7], John Vogel[2,7], Robert B. Hutmacher[8], Jeffery A. Dahlberg [4], Devin Coleman-Derr[2,9], Peggy G. Lemaux[2] & John W. Taylor [2 ✉]

Plant response to drought stress involves fungi and bacteria that live on and in plants and in the rhizosphere, yet the stability of these myco- and micro-biomes remains poorly understood. We investigate the resistance and resilience of fungi and bacteria to drought in an agricultural system using both community composition and microbial associations. Here we show that tests of the fundamental hypotheses that fungi, as compared to bacteria, are (i) more resistant to drought stress but (ii) less resilient when rewetting relieves the stress, found robust support at the level of community composition. Results were more complex using all-correlations and co-occurrence networks. In general, drought disrupts microbial networks based on significant positive correlations among bacteria, among fungi, and between bacteria and fungi. Surprisingly, co-occurrence networks among functional guilds of rhizosphere fungi and leaf bacteria were strengthened by drought, and the same was seen for networks involving arbuscular mycorrhizal fungi in the rhizosphere. We also found support for the stress gradient hypothesis because drought increased the relative frequency of positive correlations.

[1] State Key Laboratory of Mycology, Institute of Microbiology, Chinese Academy of Sciences, Beijing 100101, China. [2] Department of Plant and Microbial Biology, University of California, Berkeley, CA 94720, USA. [3] State Key Laboratory of Plant Physiology and Biochemistry, College of Biological Sciences, China Agricultural University, Beijing 100193, China. [4] University of California Kearney Agricultural Research & Extension Center, Parlier, CA 93648, USA. [5] CAS Key Laboratory of Pathogenic Microbiology and Immunology, Institute of Microbiology, Chinese Academy of Sciences, Beijing 100101, China. [6] Department of Statistics, University of California, Berkeley, CA 94720, USA. [7] Department of Energy Joint Genome Institute, 1 Cyclotron Rd., Berkeley, CA 94720, USA. [8] UC Davis Department of Plant Sciences, University of California West Side Research & Extension Center, Five Points, CA 93624, US. [9] Plant Gene Expression Center, US Department of Agriculture-Agricultural Research Service, Albany, CA 94710, USA. [10] These authors contributed equally: Cheng Gao, Ling Xu. ✉email: gaoc@im.ac.cn; jtaylor@berkeley.edu

How plants respond to drought is of profound concern because drought decreases the yields of crops that sustain human civilization[1,2]. Disturbingly, the severity and frequency of drought is increasing in many parts of the world[3–5]. In the interior and surface of different compartments such as leaf, root and rhizosphere, crop plants form essential beneficial partnerships with microbes, both fungi and bacteria, that impact plant drought responses[6]. Known benefits include the exchange of energy in the form of sugars and fats from plants for minerals acquired by microbes, notably, phosphorous from fungi and iron from bacteria[7,8]. There is no more profound change in plant transcription in response to drought than the downregulation of genes involved in managing microbial associations, and the downregulation correlates with a reduction in abundance of root-associated microbes[9]. When water again is made available, both plant transcription and microbial abundances rebound to pre-drought levels[9]. These recent insights indicate that plant drought response involves not only the plant[9], but also communities of root-associated fungal mycobiomes and bacterial microbiomes[10,11]. Therefore, our efforts to breed or engineer plants to meet the challenges of global change can be augmented by learning how to manipulate mycobiomes and microbiomes to address the crises of energy, resources, and hunger[12–14]. Of course, achieving these goals requires that we understand the plant-microbe system and how it maintains stability as it responds to stress and rebounds when stress is relieved.

Ecosystem stability in the face of disruption comprises two components: resistance and resilience[15]. Resistance is the degree to which a community remains unchanged by a disturbance, and resilience is the rate at which a community returns to its original status after being disturbed[16,17]. Resistance and resilience to stress of microbial communities were reviewed by Shade and colleagues from 247 studies across ecosystems ranging from terrestrial and aquatic natural landscapes to water treatment plants and the human gut[17]. Microbial resistance and resilience to stress were reported in ~20% of studies, almost all of which focused on bacteria[17]. In a subsequent review, studies of fungi were added to those of bacteria and consideration was given to microbial growth rate (faster for bacteria than for fungi) to form two, central hypotheses: $H_1$, that fungi should be more resistant than bacteria to drought stress, and $H_2$, that fungi should be less resilient than bacteria to rewetting[18].

Several efforts have been made to explore and test these two hypotheses from the perspective of community composition. We surveyed the literature for research that addressed community composition shifts, for both fungi and bacteria, in response to drought and subsequent rewetting (Supplementary Table 1). We find that $H_1$ has been both supported[19,20] and refuted[18,21,22], while $H_2$ has either been refuted[18,19] or remains untested[20]. These inconsistencies across studies are likely due to several factors, including differences in the methods employed to identify microbes [phospholipid fatty acids vs. rDNA metabarcoding using bacterial 16 S or fungal internal transcribed spacer (ITS)], the ecosystem type (mesocosm vs. farmland vs. grassland), the diversity of plants studied, as well as the method of imposing and relieving drought (rain exclusion vs. artificial irrigation into natural dry landscape vs. natural desiccation)[18–22]. We were particularly interested in studies of crops in agricultural fields where microbes are identified by rDNA sequence, but our survey revealed that, although some studies of crop plants used greenhouses[18,23,24] and others used metabarcoding of rDNA regions to characterize microbial communities[19–22], none investigated the resistance-resilience framework of crop plants in the field while using rDNA variation to identify microbes. Moreover, soils were the focus of these studies[18–22], leaving unexamined the hypotheses about fungal and bacterial resistance and resilience in leaf, root, and rhizosphere. Here, we examine hypotheses $H_1$ and $H_2$ for microbial communities associated with sorghum leaf, root, rhizosphere, and soil, in naturally droughted, agricultural fields experiencing two irrigation treatments, (1) regular wetting throughout the season as a control, and (2) natural, preflowering drought followed by regular wetting beginning at flowering[9–11].

Ecosystem stability relies on not only the composition of community members, but also on associations that may occur among co-existing members of microbial communities[19]. These associations range from negative to positive, from weak to strong, from non-significant to significant, and from bacterial-bacterial (B-B) and fungal-fungal (F-F) to bacterial-fungal (B-F)[19]. We make use of all these correlations to again examine $H_1$ and $H_2$ following the lead of several previous studies. These studies demonstrated that the percentage of positive correlations is related to ecological factors that include succession, fertilization, and habitat[25–31]. For example, it has been proposed that positive microbial associations should increase in frequency under stress scenarios, such as drought, a response explained by the stress gradient hypothesis (SGH)[32–37]. Thus, when microbial correlations among and between bacteria and fungi (all-correlation, B-B, F-F, B-F) are considered under the SGH, for $H_1$ (fungi are more resistant to drought stress than bacteria), drought would be expected to increase the proportion of positive correlations more strongly for B-B correlations than F-F correlations, and for $H_2$ (fungi are less resilient to rewetting than bacteria), rewetting would be expected to decrease the proportion of positive correlations more strongly for B-B correlations than F-F correlations. Although the original $H_1$ and $H_2$ were based on bacteria or fungi, by themselves, and not associations between bacteria and fungi, associations between bacteria and fungi were included in two more recent studies. First, stress studies of microbes on *Arabidopsis* leaves, roots, and the surrounding soils indicated that within-taxonomic group microbial associations tended to be positive, while those between-taxonomic groups were negative[38,39]. Second, ecological modeling indicated that negative associations should promote stability of communities[40]. Therefore, using these studies to frame hypotheses focusing on all-correlations, for our resistance hypothesis, $H_1$, under drought we expect an increase in the proportion of positive correlation most strongly for B-B, followed by F-F, and lastly by B-F correlation; and for our resilience hypothesis, $H_2$, under rewetting, we expect a decrease in the proportion of positive correlation most strongly for B-B, followed by F-F, and lastly by B-F correlation.

Co-occurrence network analysis focuses on the co-oscillation of microbial taxa in response to perturbation[19]. That is, it focuses on just the significant, positive associations. This effect has been demonstrated in mesocosms populated with various proportions of four plant species, where microbial communities were characterized via metabarcoding of bacterial 16 S rDNA and fungal ITS2[19]. Applying co-occurrence network analysis to 288 soil samples taken from four time points during periods of rain exclusion and rewetting of mesocosms, these researchers showed that drought destabilized the soil bacterial co-occurrence network, but not the fungal co-occurrence network[19]. The disruption of bacterial network by drought was also demonstrated along a transect from desert to desert grassland to typical mesic grassland[41], and along two transects of water availability that ranged from arid to hyperarid[42]. In a subtropical forest, manipulative alteration of precipitation seasonality was found to enhance fungal but not bacterial co-occurrences[43]. Therefore, when limiting consideration of resistance $H_1$ to networks of significantly positive associations, we expect that drought disruption of microbial co-occurrence network will be strongest for B-B, followed by F-F, and lastly by B-F; and, regarding resilience, $H_2$, that rewetting will strengthen microbial co-occurrence

networks most strongly for B-B, followed by F-F, and last by B-F network.

Should we expect that the microbiomes and mycobiomes that inhabit the different plant compartments (leaf, root, rhizosphere, and soil) will respond similarly to drought? Existing literature does not answer this question because previous investigations of co-occurrence networks are largely limited in one compartment (Supplementary Table 1). By considering all four compartments in previous reports, we showed that drought responses of fungal and bacterial communities are most pronounced in root, followed by rhizosphere and, lastly, soil and leaves, where the responses were much weaker[10,11] (Fig. 1). Guided by these results, here we extend the network hypothesis to all four plant compartments: drought disrupts microbial network more strongly in root than rhizosphere, soil and leaf compartments.

Identification of key network elements, in this case modules or hubs, may facilitate practical application of microbial networks to modern agriculture. Modules, the highly interconnected sub-structures within networks, may represent ecological units comprising highly interacting members[44]. Microbes with the most links to other microbes occupy central positions in modules and are termed modular hubs. Microbes with links to microbes in other modules are termed connectors. The connectors with the most links to other modules occupy the center of networks and are termed network hubs, which may also be modular hubs in their home module[45]. Network hubs and modular hubs are both disproportionally important in structuring microbial communities[39]. Artificial inoculation of these hub taxa might provide a means of directing the microbial community, or key modules within the community, to reduce inputs or improve yields for modern agriculture[46].

Here, we address the hypotheses about resistance ($H_1$) and resilience ($H_2$) using three approaches, (i) whole community composition, (ii) all pairwise correlations among individual taxa, and (iii) the co-occurrence network of significant positive associations. In a semiarid agricultural field where control plots were watered regularly and test plots were naturally droughted before flowering followed by regular wetting beginning at flowering[9–11], we used modern high-throughput sequencing techniques to examine communities of bacteria and fungi associated with leaf, root, rhizosphere, and surrounding soil of two sorghum cultivars planted as a monoculture during a growing season. One might wonder if the microbes in these fields were already adapted to drought, however a six-decade history of irrigated agriculture at the site indicates that the microbes in our system are not drought-adapted[11]. Thus, this experimental system allowed us to investigate resistance and resilience of these microbial communities under the stress of drought and subsequent recovery after watering. Community assembly of both fungal mycobiome and bacterial microbiome were published earlier in separate papers[10,11]. Here, we newly analyzed these two datasets together to test $H_1$ and $H_2$ using the three approaches noted above. At the level of community composition, our results show that fungi, as compared to bacteria, are more resistant to drought stress but less resilient when drought is relieved by rewetting, although stronger resilience of the fungal than the bacterial community is observed in the first week of rewetting. Furthermore, although drought generally disrupts microbial networks, co-occurrence networks among functional guilds of rhizosphere fungi and leaf bacteria are strengthened by drought.

## Results

**Testing $H_1$ and $H_2$ at community composition level**. As noted above, the simple fact that fungi grow more slowly than bacteria is the basis of the hypotheses that ($H_1$) fungal communities should be more resistant than bacterial communities to drought stress, and ($H_2$) that fungal communities should be less resilient than bacterial communities when the stress is relieved by rewetting[18]. In addition to growth rate, these two hypotheses may be related to differences in the form of growth between fungi and bacteria. For example, multicellular hyphal growth versus unicellular division or the greater thickness of fungal cell walls as compared to those of bacteria[47,48]. We tested $H_1$ and $H_2$ at the community composition level by blending the fungal and bacterial datasets generated from the same leaf, root, rhizosphere and soil samples collected from field-grown sorghum that had been either irrigated as a control, or subjected to preflowering drought followed by regular wetting beginning at flowering[10,11].

We followed the approach of Shade et al.[17] to detect resistance and resilience, which had been developed for univariate variables, e.g., richness. For multivariate data, e.g., community composition, we modified it by calculating pairwise community dissimilarity for two groups: within-group (control-control pairs, drought-drought pairs, or rewetting-rewetting pairs), and between-group (control-drought pairs, or control-rewetting pairs). Ecological resistance to drought stress is detected by comparing compositional dissimilarity of between-group pairs (control-drought pairs) against within-group pairs (control-control pairs and drought-drought pairs) for each of the droughted weeks (weeks 3–8). Ecological resilience to rewetting is detected by assessing, from before to after rewetting, the change in the difference of compositional dissimilarity between within-group pairs and between-group pairs. Here, the point just before rewetting was week 8 and the points after rewetting were weeks 9–17. A t-test was used to assess the statistical significance of the differences in resistance or resilience between bacterial and fungal communities at each time point for each compartment.

To account for the different resolutions of ITS and 16 S, we compared bacterial 16 S OTUs against both fungal ITS, species-level OTUs as well the fungal family level (Supplementary Fig. 1). The results of analyses using either fungal families or OTUs are consistent. Out of 36 comparisons (15 roots, 15 rhizospheres and 6 soils), different family and OTUs results were detected in four instances. In two of these, significances detected by OTUs were not detected by family (root, weeks 4 and 17) and, in the other two cases, significances detected by family were not detected by OTUs (rhizosphere, weeks 7 and 8). (Fig. 1). We report only results that are consistent at both the species and family levels (Fig. 1).

In line with our first hypothesis, $H_1$, we found that the resistance to drought stress for fungal mycobiomes was consistently stronger than that for bacterial microbiomes for weeks 5 in root, weeks 4–6 in rhizosphere, and weeks 4 and 6–8 in soil (Fig. 1, Supplementary Fig. 1 and Supplementary Table 2). In support of our second hypothesis, $H_2$, when the stress of pre-flowering drought was relieved by rewetting, we found that the resilience of the bacterial communities was consistently higher than that for the fungi in weeks 9–16 in root, and weeks 11–17 in rhizosphere (Fig. 1, Supplementary Fig. 1 and Supplementary Table 2).

Surprisingly, we found that resilience was stronger for fungal than bacterial communities in the first week (week 9) of rewetting in the rhizosphere (Fig. 1, Supplementary Fig. 1 and Supplementary Table 2). This high resilience of fungi may be associated with the quick growth of sorghum roots when rewetted. The rhizosphere zone around these newly formed roots may be quickly colonized by soil fungi, a community that was weakly affected by drought. This result suggests that re-assembly of the rhizosphere microbial community is more complex than previously expected.

The finding that fungal community composition in the soil is not shaped by drought prevented us from further detecting

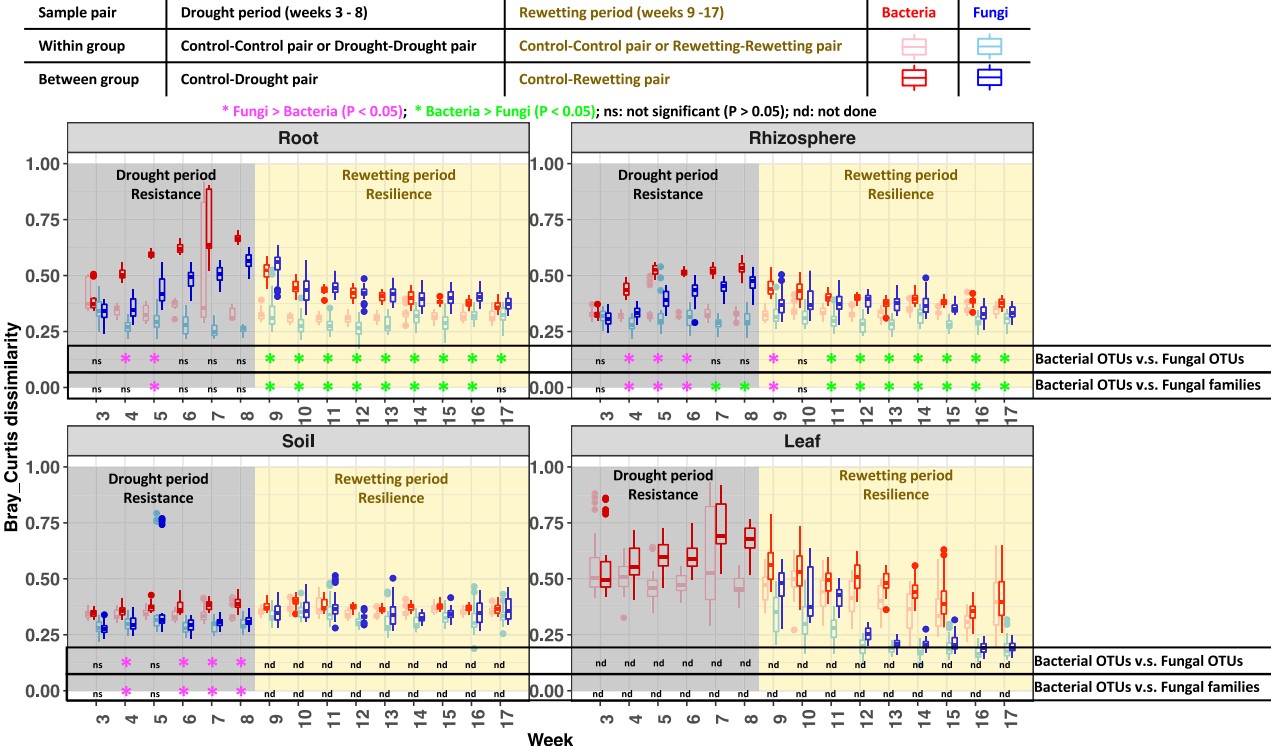

**Fig. 1 Resistance and resilience of bacterial and fungal community composition.** Bray-Curtis dissimilarities were computed for bacterial (red and light red) and fungal (blue and light blue) communities of four compartments (root, rhizosphere, soil, leaf) of $n = 12$ biologically independent plots examined over 17 weeks. The boxes represent the 25th–75th percentiles (with the median as a horizontal line) and the whiskers show the 10th–90th percentiles. Ecological resistance to drought stress is detected by comparing compositional dissimilarity of between-group pairs (control-drought pairs) against within-group pairs (control-control pairs and drought-drought pairs) at each of the droughted weeks (weeks 3–8, the grey shaded area) (*$p < 0.05$, adjusted by Bonferroni method; unpaired t-test, two-sided). Ecological resilience to rewetting is detected by assessing, from before, to after, rewetting, the change in the difference of compositional dissimilarity between within-group pairs and between-group pairs. Here, the point just before rewetting was week 8 and the points after rewetting were weeks 9–17 (the gold shaded area). To account for the different resolution of internal transcribed spacer (ITS) and 16 S, we compared bacterial 16 S operational taxonomic units (OTUs) against both fungal ITS OTUs, as well as fungal families. In 32 of 36 communities, the results of fungal families and OTUs are consistent. Different family and OTUs results were detected in two cases where significances detected by OTUs were not detected by family (root, weeks 4 and 17), and in two cases where significances detected by family were not detected by OTUs (rhizosphere, weeks 7 and 8). We report only results that are robust across these two conditions. Significantly higher resistance to drought of fungi than bacteria was detected in root (week 5), rhizosphere (weeks 4–6) and soil (weeks 4, 6–8). Significantly higher resilience to rewetting of bacteria than fungi was detected in root (weeks 9–16) and rhizosphere (weeks 11–17). Note that fungi exhibited stronger resilience than bacteria at the first week of rewetting (week 9). The finding that fungal community composition in soil is not shaped by drought prevented us from further detecting resilience in this compartment. Note that fungal communities in early leaves are excluded from analysis due to the high proportion of non-fungal sequencing reads. The detailed results at fungal family levels can be found in Supplementary Fig. 1. Source data are provided as a Source Data file.

resilience (Fig. 1). Note fungal community in early leaves was excluded from analysis due to the high proportion of non-fungal reads in sequencing[11].

**Testing $H_1$ and $H_2$ at all-correlation level.** Next, we moved from the comparison of whole communities to correlation among individual bacterial and fungal taxa to test the hypotheses about resistance, $H_1$, and resilience, $H_2$. As noted above, previous research provided the foundation for the stress gradient hypothesis, which predicts an increase in positive associations in stress[32–37]. Further, ecological modeling predicts that negative associations promote stability[40]. Concerning specific associations, studies of *Arabidopsis* and associated microbes reported that positive associations are favored within kingdoms, i.e., within bacteria or within fungi, while negative associations predominate between kingdoms[38,39]. Given these foundations, concerning $H_1$, we expected an increase in the proportion of positive correlation by drought stress that would be strongest for B-B, followed by

F-F, and lastly by B-F; for $H_2$ we expected rewetting to cause a decrease in the proportion of positive correlation, again most strongly for B-B, followed by F-F, and lastly by B-F.

Overall, at the all-correlation level, we found no consistent support for the differences postulated for bacterial and fungal responses in $H_1$. For example, strong increases in the proportion of positive correlations under drought could be found in all microbial pairings for some compartments (B-B in leaf and root, F-F in rhizosphere and soil, and B-F in root and rhizosphere) (Fig. 2a, Supplementary Figs. 2, 3). Neither did we find consistent support for the differences ascribed to bacteria and fungi in $H_2$ as the strongest decreases in the proportion of positive correlations during rewetting occurred at F-F in rhizosphere and soil, and B-B in leaf and root (Fig. 2b, Supplementary Figs. 2, 3).

We found support for the stress gradient hypothesis because drought increased the relative frequency of positive correlations among microbial taxa (Fig. 2a, Supplementary Figs. 2, 3). The increases were due, largely, to B-B correlations in leaf and root F-F correlations in the rhizosphere during drought, when the relative

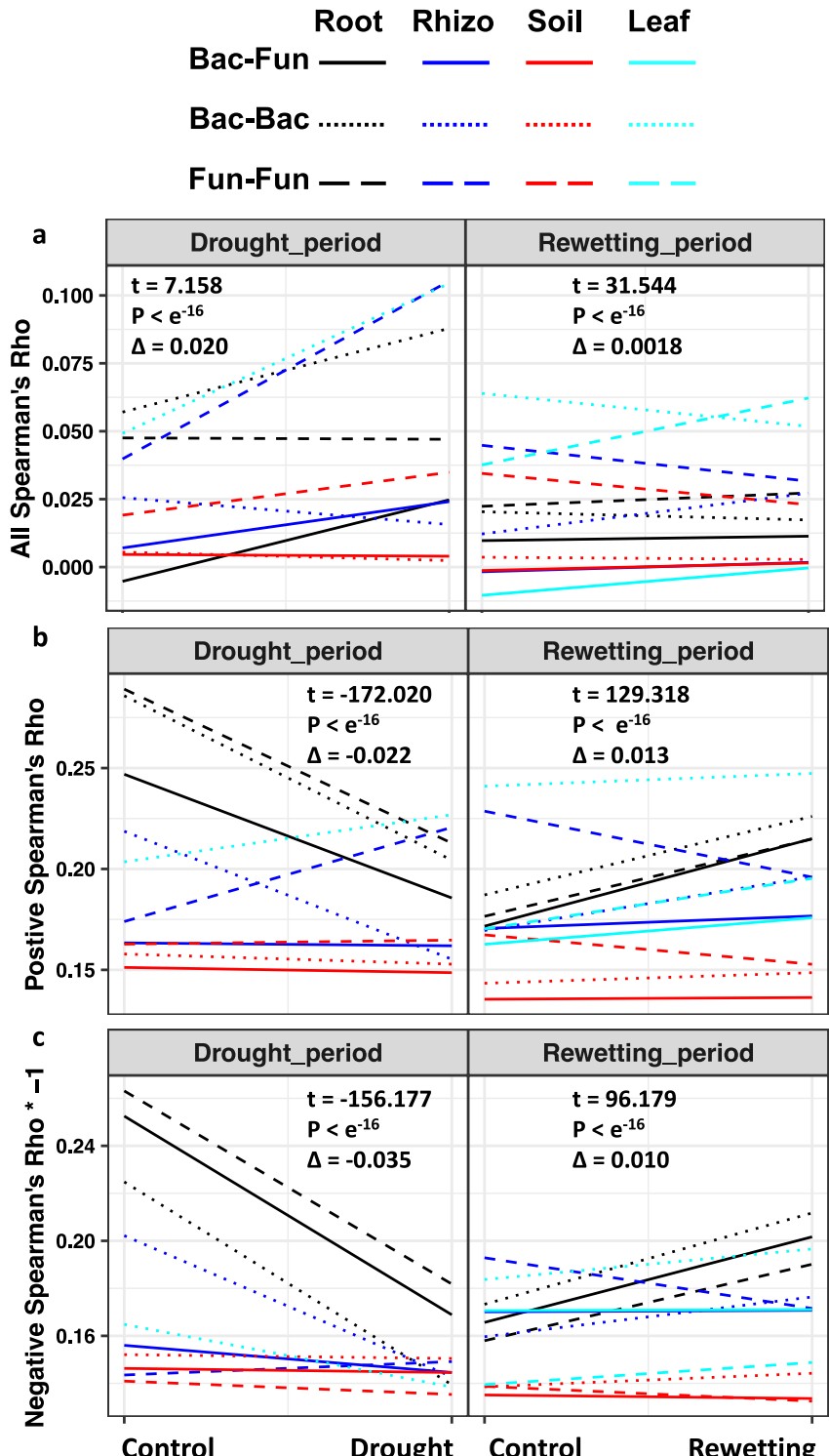

**Fig. 2 Correlations of microbes in drought stress and drought relief.** Estimates of combined correlations (row a) show an increase in positive correlations under drought stress across the four compartments (root, black; rhizosphere, blue; soil, red; leaf, green). Data points underlying the lines in the figure are provided in the alternative version in Supplementary Fig. 2. This result is in line with the stress gradient hypothesis which posits that stressful environments favor positive associations because competition will be less intense than in benign environments[32,33,36,37]. Note that positive trends in combined correlations can arise in two ways. First, from an increase of positive correlations (row b) that exceeds the rise in negative correlations (row c), e.g., Leaf bacterial-bacterial (Bac-Bac) correlations or rhizosphere fungal-fungal (Fun-Fun) correlations in the drought period (Negative correlations in row C values are multiplied by −1 to facilitate comparison). Second, from a decrease in negative correlations that exceeds a decrease in positive correlations, e.g., root bacterial-bacterial correlations or root bacterial-fungal (Bac-Fun) correlations in drought. Combined (a), positive (b) and negative (c) estimates of correlation (Spearman's rho, $\rho$) are given for four compartments (root, rhizosphere, soil and leaf), and three types of correlations (Bacterium-Bacterium, Fungus-Fungus, Bacterium-Fungus). T-tests (two sided) were carried out for linear mixed effect modelling that incorporates link type and compartments as random factors. Detailed distribution densities of correlations are presented in Supplementary Fig. 3. Source data are provided as a Source Data file.

frequency of positive correlations was increased (Fig. 2b, Supplementary Figs. 2, 3) and the frequencies of negative correlations were decreased or weakly increased (Fig. 2c, Supplementary Figs. 2, 3). Less obvious increases in the relative frequency of positive correlations (such as B-B in root, F-F in soil, and B-F in root and rhizosphere) occurred where drought reduced both positive and negative correlations, but the losses of negative correlations exceeded those of positive correlations (Fig. 2, Supplementary Figs. 2, 3).

In support of the expectation that correlations would be more negative between taxonomic groups than within taxonomic groups, we found that the relative frequency of positive correlations was generally lower for B-F than B-B and F-F correlations (Fig. 2, Supplementary Figs. 2, 3). Moreover, as ecological modeling has indicated that negative associations should promote stability of communities[40], we hypothesize that B-F correlations would be more stable than B-B and F-F networks in response to drought stress. However, we found no support for this hypothesis, as B-F correlations (for example in root) did not always show the least response to drought stress (Fig. 2, Supplementary Figs. 2, 3).

**Testing H₁ and H₂ at species co-occurrence level**. For our final test of $H_1$ (resistance) and $H_2$ (resilience) we focused on co-occurrence networks based on significant, positive correlations. These networks have been reported to be destabilized for bacteria but not for fungi in mesocosms subject to drought stress[19], and shown to be disrupted for bacteria in natural vegetation studied over gradients of increasing aridity[41,42]. Using these results as guides, for $H_1$ we expected that drought stress should disrupt co-occurrence networks most strongly for B-B, followed by F-F, and lastly by B-F. For $H_2$ we expected that relief of stress by rewetting should strengthen microbial co-occurrence networks most strongly for B-B, followed by F-F, and lastly by B-F.

For this test we constructed microbial co-occurrence networks using significant positive pairwise correlations between microbial taxa, B-B, F-F and B-F, and compared the network complexity between fully irrigated control and drought, and between control and rewetting following drought. In general, we found no consistent support for the difference between bacteria and fungi inherent in $H_1$. Rhizosphere was the one compartment where B-B vertices dropped and F-F vertices rose in response to drought, as expected, but this result was offset in root and soil, where vertices dropped in all networks, B-B, F-F and B-F (Figs. 3, 4; Supplementary Figs. 4, 5). Analysis by co-occurrence networks highlighted the differences between plant compartments. In root drought strongly disrupted networks of B-B, B-F and F-F, but in the other three compartments, network disruption was weaker, and networks were even enhanced by drought for F-F in rhizosphere and B-B in leaf (Figs. 3, 4).

Our second hypothesis $H_2$ was supported by comparison of rewetted and control microbial networks in the rhizosphere and soil where F-F networks became less complex and B-B networks became more complex, again likely due to the slower growth rate of fungi than bacteria. Behavior of the B-F network largely followed the patterns of B-B in root, rhizosphere and soil (Figs. 3, 4, Supplementary Figs. 4, 5). However, we found no support for the $H_2$ in leaf and root where the F-F did not lose complexity, and where both the B-B and B-F networks gained complexity (Figs. 3, 4, Supplementary Figs. 4, 5). The results in root and leaf indicate that, upon rewetting, the co-occurrence network disrupted by drought quickly strengthens to become even more complex than the undisturbed control. Our results suggest that resilience does not necessarily stop when approaching the control values, but that resilience of biotic association can exceed the

control. Our data highlight a phenomenon that has rarely been reported[17].

We then turned our attention to detecting network modules, finding that network modularity is generally increased by drought stress, and decreased by rewetting (Supplementary Figs. 6–9). The exceptions were the F-F network in rhizosphere and the B-B network in leaf, both of which showed lower modularity under drought (network modularity, F-F: 0.483; B-B: 0.600) than control (network modularity, F-F: 0.698; B-B: 0.835), and higher modularity in rewetting (network modularity F-F: 0.529; B-B: 0.314) than control (network modularity F-F: 0.390; B-B: 0.247) (Supplementary Figs. 6–9).

We then detected the hub taxa of the networks based on their links to other microbes within modules (modular hubs, Zi) and connector taxa based on their links to other microbes in other modules (connectors, Pi). Interestingly, recalling the observed decrease during drought of modularity of the networks for rhizosphere F-F and leaf B-B, we found that the numbers of connectors of both networks were higher for drought than control (Supplementary Figs. 10–13), indicating that higher modularity can result in fewer, larger modules and fewer opportunities for connectors. Specifically, in the rhizosphere F-F network, four connectors [arbuscular mycorrhizal OTU70_-Claroideoglomus (Pi = 0.643); saprotroph OTU93_Mortierella (Pi = 0.625); plant pathogens OTU87_Spizellomyces (Pi = 0.64) and OTU624_Cylindrocarpon (Pi = 0.667)] and one modular hub (saprotroph OTU59_Chaetomium, Zi = 2.54) were detected under drought, whereas no network hub, module hub or connector was detected in controls (Supplementary Fig. 11; Supplementary Data 1). In the leaf B-B network, five connectors [four Actinobacteria (Pi = 0.625–0.678) and one Chloroflexi (Pi = 0.667)] and three module hubs [two Actinobacteria (Zi = 3.254) and one Proteobacteria (Zi = 2.852)] were detected under drought, whereas no network hub, module hub or connector was detected in controls (Supplementary Fig. 12; Supplementary Data 1). The decrease of modularity and increase of hub numbers indicate that rhizosphere fungi and leaf bacteria are more interconnected under drought (Fig. 3b).

**Guilds**. The strong response to drought stress demonstrated by rhizosphere fungi and leaf bacteria encouraged us to sort fungi and bacteria into functional groups, guilds for fungi and phyla for bacteria. When fungi or bacteria, alone, were displayed in networks, the gain in complexity in stress was more apparent (Fig. 3b) and it became possible, for the gain, to calculate the proportion of new correlations that were within or between guilds or phyla (Fig. 3c). For rhizosphere fungi, the inter-guild correlations formed during rewetting were higher than those of constantly watered controls, as was the case with leaf bacteria where the distribution of inter-phyla correlations formed during rewetting was higher than control (Fig. 3c).

Fungi belonging to one of the guilds, arbuscular mycorrhizal fungi (AMF), form key mutualistic symbiosis with plants, and interact with soil microbiome to contribute to the host plant's adaption to various biotic and abiotic stresses[49]. Here we explored the resistance and resilience of significant, positive, co-occurrences between AMFs and other fungi, and between AMFs and bacteria. We found that the network of AMF and other fungi was disrupted in root and soil but was strengthened in rhizosphere, and that the network of AMF and bacteria was disrupted in root, rhizosphere, and soil. Networks in roots and soil of both AMF and other fungi and AMF and bacteria, when re-wetted, largely recovered their pre-drought complexity. In rhizosphere, however, the network of AMF and other fungi was less complex in rewetting than the control (Fig. 5a), and the

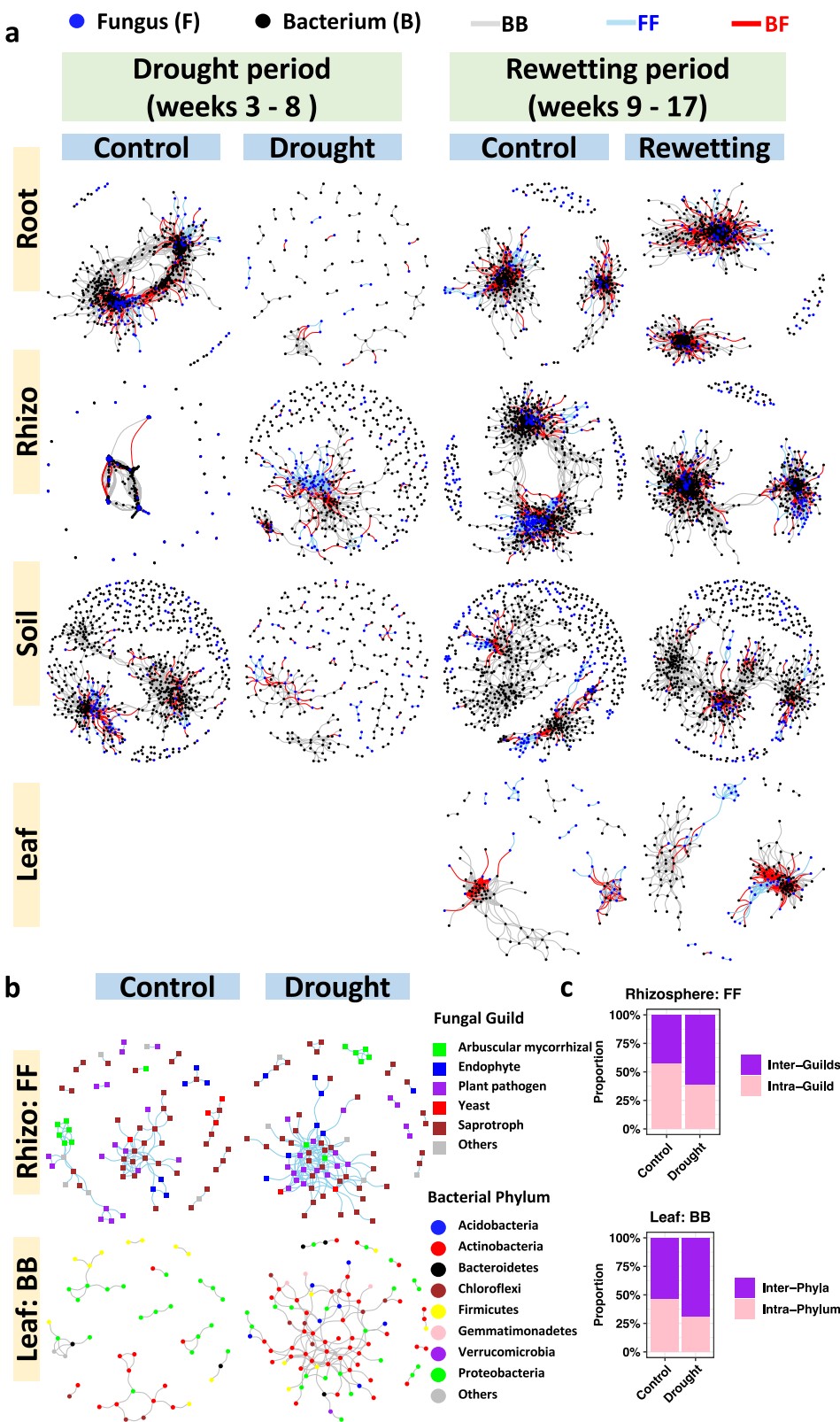

network of AMF and bacteria was not different from the control (Fig. 5b).

## Discussion

Our analyses show that inferences made about bacteria and fungi based on community composition change when co-occurrence, which implies association, is added to the analyses. Our application of microbial community comparison supported both $H_1$ that fungi are more resistant than bacteria to drought stress and $H_2$ that fungi are less resilient than bacteria to rewetting, which is not surprising because these hypotheses were formulated from community studies and have found partial support in other community studies, e.g. de Vries, et al.[19] and Barnard, et al.[20].

**Fig. 3 Networks of significant positive cross-taxonomic group correlations (bacteria and fungi). a** Fungal operational taxonomic units (OTUs) (blue) and bacterial OTUs (black) are graphed as nodes. Significant positive Spearman correlations are graphed as edges ($\rho > 0.6$, false discovery rate adjusted $P < 0.05$, two-sided); Skyblue (fungus-fungus, FF), grey (bacterium-bacterium, BB) and red (bacterium-fungus, BF). All three types of co-occurrences (BB, FF and BF) are generally disrupted by drought (but not FF in rhizosphere and BB in leaf, see Supplementary Fig. 3), and recovered by rewetting. **b**, **c** FF co-occurrences in rhizosphere and BB co-occurrences in leaf are enhanced by drought, which is coupled with the increase of the proportion of associations between fungal guilds and the increase of the proportion of associations between bacterial phyla. The key finding that drought enhanced the rhizosphere, fungal network and the leaf, bacterial network was also supported by the Pearson and CoDa methods (Supplementary Figs. 18, 19). Source data are provided as a Source Data file.

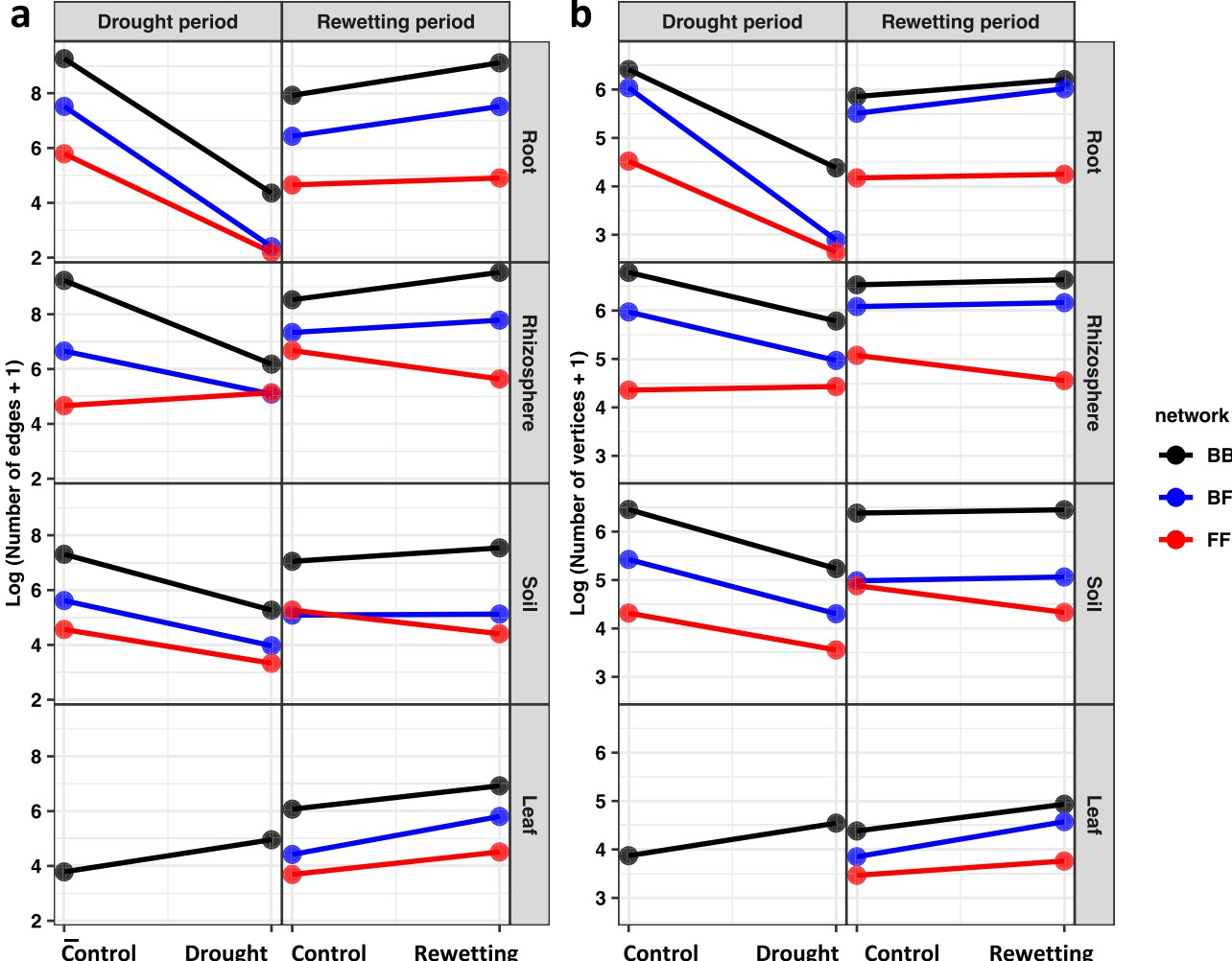

**Fig. 4 Microbial co-occurrences in drought-stress and drought relief.** The complexities of microbial co-occurrence networks are demonstrated by **a** the number of vertices and **b** the number of edges. All three types of co-occurrences (fungus-fungus, FF, red; bacterium-bacterium, BB, black; and bacterium-fungus, BF, blue) are generally disrupted by drought (but not FF in rhizosphere and BB in leaf) and recovered by rewetting. Source data are provided as a Source Data file.

However, even restricting analyses to community composition, support for these two hypotheses is not universal; for example, $H_1$ and $H_2$ were rejected by de Vries and Shade[18], $H_1$ was rejected by Mc Hugh et al.[22] and McHugh and Schwartz[21], and $H_2$ was rejected by de Vries, et al.[19]. The main difference between our study and these others is the simplicity of our system, the use of DNA metabarcoding to identify microbes and the dependability of natural drought in an arid environment. We used just one species of plant whose growth is synchronous whereas all other studies focused on at least four species[19] and typically many plant species. We used DNA sequences of variable regions to identify bacteria and fungi. Our field season was free of precipitation for the entire growing season, making it straightforward to experience drought and then relieve it through irrigation. Another difference is our avoidance of heterogeneity of variance among different time periods (e.g., de Vries, et al.[19]) by comparing compositional dissimilarity among communities within treatments (combined control and stress) with dissimilarity between control and stress communities.

When we moved to consider all correlations to examine the two hypotheses about resistance $H_1$ and resilience $H_2$, developed from community studies, the result was complex. Rather than observing the expected differences between B-B and F-F associations, we found that drought stress increased the proportion of positive correlations for both B-B and F-F in multiple compartments (Fig. 2). Extending the analysis to previously poorly

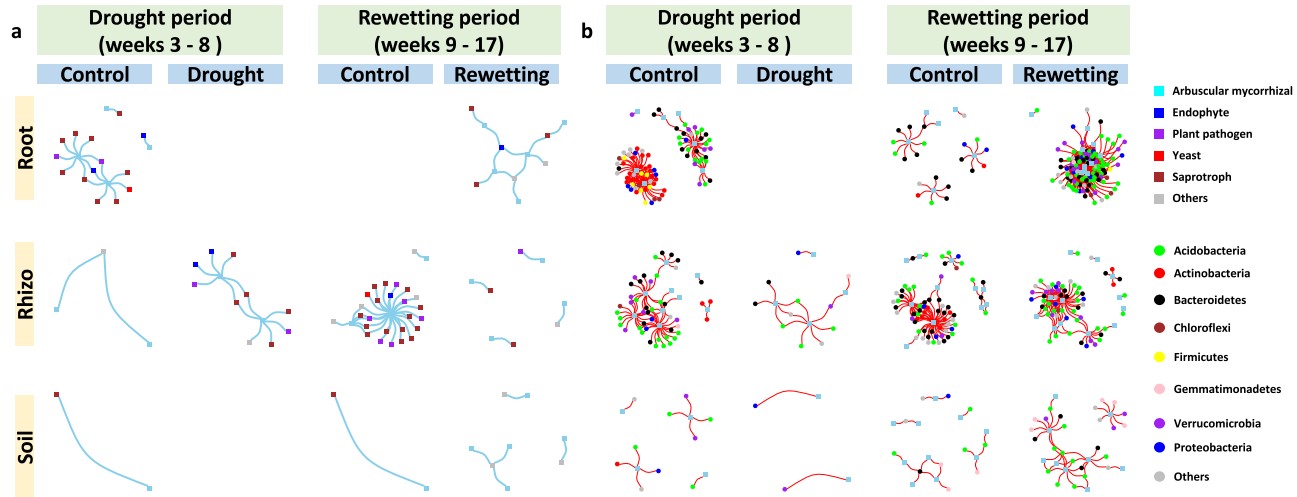

**Fig. 5 Network of co-occurrences related to arbuscular mycorrhizal fungi (AMF). a** Network of significant positive co-occurrences between AMFs and other fungi. **b** Network of significant positive co-occurrences between AMFs and bacteria. Source data are provided as a Source Data file.

examined B-F associations, we found increases in these associations in root and rhizosphere but no change in soil (Fig. 2). Similar complexity was seen in applying all-correlation analysis to $H_2$ in that the strongest decreases in the proportion of positive associations could occur either B-B or F-F (Fig. 2).

Signals of co-occurrence may be masked in all-correlation analyses that include correlations that are both positive and negative, and both nonsignificant and significant[19]. However, when we limited the analyses of our resistance hypothesis $H_1$ to networks of correlations that are both significant and positive, we found an outcome similar to that seen for all correlations–some combinations of compartment and stress showed support for $H_1$ and others did not. Despite this variation, there were some clear trends. At the dimension of plant compartments, drought disrupted root networks more strongly than those of other compartments. In almost every compartment under drought stress, B-B co-occurrences declined, with leaf in preflowering drought providing the only exception (Fig. 3). Almost the same can be said for F-F, where the only examples of increasing positive network correlations occurred in drought rhizosphere (Fig. 3). In no case did the B-F positive network correlations increase, and they declined in every compartment under drought. In short, drought disrupts microbial networks. Given that nearly all the energy that supports all of the microbes comes from the photosynthesis provided by the plant, it is not surprising that drought, which limits photosynthesis[9], should also strongly disrupt the microbial networks. This result may reflect the stronger reduction of plant resources in the root, which would lead to stronger disruptions of bacterial and fungal networks in this compartment.

Previous studies also report disruption by the drought of soil bacterial co-occurrence networks along natural arid gradients[41,42], but another study did not report any effect of drought on soil fungal co-occurrence networks in potted plants[19]. Our study of field-grown plants shows that drought can enhance as well as disrupt microbiome networks, emphasizing the positive role that bacterial and fungal communities can play in plant drought response. The strongest surprises were that drought increased bacterial co-occurrences in leaf and fungal co-occurrences in the rhizosphere. The increase in bacterial co-occurrences by drought might be related to a previous observation in this same sorghum system, that abundance of fungal yeasts, which receive nutrients by diffusion as do bacteria, increases shortly before flowering[11]. Why F-F, and not B-B or B-

F, would increase by drought in rhizosphere when all co-occurrences are declining in root and soil is more difficult to understand. Perhaps the reduction in nutrients experienced by root inhabiting fungi is not enough to discourage more oligotrophic rhizosphere fungi, or perhaps droughted roots release nutrients to the rhizosphere, either directly or as a consequence of senescence[9].

Turning to our resilience hypothesis, $H_2$, we found, as expected, that bacterial co-occurrence networks recovered strongly while rewetted fungal networks showed further disruption in rhizosphere and soil, and weak recovery in leaf and root. Our exploration of B-F co-occurrences shows a gain in positive associations in all compartments, except soil. Apparently, restoration of water, which leads to restoration of plant photosynthesis[9], brings back the disrupted microbial communities most reliably in leaf and root, where newly produced photosynthate would be most available, and less so in rhizosphere or soil. These results could also be explained by a slower response in the rhizosphere or soil that was not captured over the period of our study.

Sorting the microbes into fungal guilds or bacterial phyla allows us to speculate about ecological function. Both the rhizosphere fungal network and the leaf bacterial network strengthened in drought stress. For fungi, the increase in network association was coupled with an increase of fungal inter-guild co-occurrences. For bacteria, the increase in network association was accompanied by an increase in inter-phylum co-occurrences (Fig. 3c). These results suggest that the strengthening of co-occurrences might be underpinned by niche differentiation and functional complementarity among taxa. Note the strengthening of fungal networks in rhizosphere was coupled with a drastic decrease of fungal richness[11]. Given that the microbial network should reflect function[50,51], the loss of rhizosphere fungal diversity must imply a loss of potential ecosystem functioning. The strengthened fungal network in the rhizosphere seen in this study was coupled with the co-occurrence of a number of fungal pathogens with saprotrophic, endophytic and mycorrhizal fungi. However, it is not likely that there was an increase in plant decay or disease, because we previously found that the relative abundance of rhizosphere fungal pathogens was drastically decreased by pre-flowering drought[11]. Still, the question remains, why is network complexity rescued only for bacteria in leaf, and only for fungi in rhizosphere? Also, it's unclear whether a different pattern

would be observed if the micro- and mycobiomes were investigated over longer periods.

Moving from comparison of entire communities to comparison of all possible correlations between pairs of microbes, we examined several hypotheses developed from empirical studies. Our analyses supported the stress gradient hypothesis[31–37] in that positive associations increased overall in the stresses of drought, led by abundant B-B associations in leaves and F-F associations in the rhizosphere (Fig. 2). We also found support for the hypothesis that associations would be less negative within fungi or within bacteria than between the two taxonomic groups[38,39] (Fig. 2). We also tested a hypothesis that emerged from modeling, finding no support for the concept that negative associations would support stability[40] (Fig. 2).

The detected associations in networks may be composed of a mixture of real and false interactions, of direct and indirect interactions, and of physical and chemical interactions. However, we note that correlation does not necessarily equate with interaction, but also can be ascribed to habitat-filtering, niche sharing or dispersal limitation[52]. As is the case with most field-based experimental designs, it is not possible to assess the effect of habitat filtering and niche sharing. However, we can note that the role of dispersal limitation on the co-occurrence network is weak. Based on our implementation of a taxon-taxon-space association approach, the percentage of network links related to spatial distance was no more than three percent (0–2.94%; Supplementary Fig. 14). This result echoes the absence of a significant relationship between spatial distance and dissimilarity of microbial community composition reported in our previous study[11]. Thus, dispersal limitation is not likely the driver of microbial association and community composition in our small research site (~500 m²), which has been cultivated for nearly six decades and was planted to one crop (sorghum) throughout our study[11].

While the exact nature of correlative associations cannot be recognized by our amplicon-based method, the changes in network complexity and detections of network hubs can be used to infer ecological function. For example, our application of co-occurrence analyses demonstrated the dynamic nature of microbial communities in response to drought and suggested possible strategies to use microbes to improve plant drought tolerance. In terms of translating basic research to agricultural practice, the strengthening in the drought of fungal networks in the rhizosphere and bacterial networks in leaves are prime targets for microbiome engineering (Fig. 3b, Supplementary Fig. 4). Given that microbial networks show association with function[50,51], the drought-strengthened networks may help the host plant adapt to drought. This association suggests that inoculation of the hub taxa might rescue the drought-disrupted networks and improve drought tolerance. For example, in systems where the F-F network is disrupted by drought stress, the rhizosphere F-F network might be rescued by artificial inoculation of the arbuscular mycorrhizal OTU70_Claroideoglomus and saprotrophic OTU93_Mortierella and OTU59_Chaetomium, the three hubs of F-F network that we detected under drought stress (Supplementary Data 1). Similarly, for systems where the B-B network is disrupted by drought stress, the leaf B-B network might be rescued by artificial inoculation of drought tolerant, Monoderms (Actionobacteria and Chloroflexi), members of the bacterial hubs detected under drought stress in this study (Supplementary Data 1).

## Methods

Data of the fungal mycobiome and bacterial microbiome have been published in separate papers that investigated the effect of drought on the development of bacterial and fungal communities[10,11]. The current research integrated these two datasets to test the hypotheses that fungi are (i) more resistant to drought stress but (ii) less resilient when the stress is relieved by rewetting than bacteria at levels of both community composition, and microbial associations as inferred from pairwise correlations and microbial co-occurrence networks. The experimental design, sampling, and bacterial and fungal metabarcoding analyses described here are summarized from our previous publications on research conducted at the same study site[10,11].

**Experiment design and sampling.** The experiment was conducted from 27 May to 28 September, 2016, at Kearney Agricultural Research and Extension (KARE) Center in Parlier, CA, USA (36.6008° N, 119.5109° W). No precipitation was recorded during our experiment[49]. Our experiment had three treatments (control, pre-flowering drought and post-flowering drought), two sorghum [Sorghum bicolor (L.) Moench] cultivars (RTx430 and BTx642), in three replicating plots (16 * 8 m² each) with a random block design. The finding that the effects of sorghum cultivar on both bacterial and fungal communities were negligible allowed us to use six replicates in our analyses[10,11]. The trial was planted on May 27, 2016, and plant emergence was recorded on June 1, 2016[9–11,49,53]. From the 3rd week until the 17th week of growth, the control plots were regularly watered, the pre-flowering drought treatment were not watered until the ninth week at which time regular watering was initiated, and the post-flowering drought treatment was regularly watered until the 10th week at which time watering ceased. Weekly samples of leaf, root, rhizosphere and soil were collected for control plots from June 8 (week 1) to September 28 (week 17), as detailed in our previous publications[9–11,49]. To avoid redundancy of control conditions, pre-flowering treatment sampling began on week 3 and post-flowering sampling began on week 8[9–11,49]. In total, 1026 samples were collected[11].

The experimental design of pre-flowering drought followed by regular wetting beginning at flowering represents an ideal system for testing the hypotheses that fungi are (i) more resistant to drought stress but (ii) less resilient when the stress is relieved by rewetting than bacteria. However, the experimental design of regular watering followed by postflowering drought is not relevant to these two hypotheses. Therefore, for simplicity, this study only included control and preflowering drought (followed by rewetting) treatments and did not analyze the postflowering drought treatment.

Detailed description of DNA extraction, fungal ITS2 and bacterial 16 S amplifications and MiSeq sequencing can be found in our publication of the sorghum mycobiome and microbiome[10,11]. Briefly, DNA was extracted from 0.2 g of leaf, root, rhizosphere, or soil samples using the MoBio PowerSoil DNA kit (MoBio, Carlsbad, CA, USA)[10,11]. Fungal ITS2 and bacterial 16 S were PCR-amplified from DNAs diluted to 5 ng/μl with ddH₂O, using dual-barcoded 5.8SFun (5'-AACTTTYRRCAAYGGATCWCT-3') /ITS4Fun (5'-AGCCTCCG CTTATTGATATGCTTAART-3') (fungi) and 341 F (5'-CCTACGGGNBGCAS CAG-3') /785 R (5'-GACTACNVGGGTATCTAATCC-3') primers (bacteria)[10,11]. The yields of PCR products were quantified, pooled, purified, and then sequenced on the Illumina MiSeq PE300 sequencing platform (Illumina, Inc., CA, USA) at the Vincent J. Coates Genomics Sequencing Laboratory (GSL, University of California, Berkeley, CA, USA)[11]. Raw fastq sequences were subjected to quality evaluation, removal of primers, merging of forward and reverse reads, control of quality and clustering of OTUs[10,11]. Fungal OTUs were assigned into functional guilds using the FUNGuild v1.1[54].

**Statistical methods.** To account for the difference in the number of reads, both bacterial and fungal datasets were rarefed to 10,000 read per sample using the rrarefy function in package vegan in R version 4.2.0[55]. The proportion of fungal reads was low in early leaves (weeks 1–8) due to nonspecific amplification[11], so we excluded these fungal data from our analyses. To decouple time and treatments, we split all samples into two datasets: i) weeks 3–8 of control and preflowering drought treatments aimed to evaluate the effect of preflowering drought; and ii) weeks 9–17 of control and preflowering drought treatments aimed to evaluate the effect of rewetting[56].

We followed Shade et al[17] for the detection of resistance and resilience. The original method of Shade et al[17] was developed for univariate variables such as richness. To test H₁ and H₂ at multivariate community composition level, matrices of Bray-Curtis dissimilarity between communities were calculated using the vegdist command in vegan 2.6.2[55]. From the Bray-Curtis dissimilarity matrix, subsets of dissimilarity were selected between communities within groups (control-control pairs, drought-drought pairs, or rewetting-rewetting pairs) and those between groups (control-drought pairs, or control-rewetting pairs). Ecological resistance to drought stress was detected by comparing compositional dissimilarity of between-group pairs (control-drought pairs) against within-group pairs (control-control pairs and drought-drought pairs) at each of the droughted weeks (weeks 3–8). Ecological resilience to rewetting was detected by assessing the change from the time point prior to rewetting (week 8) to each of the time points following rewetting (weeks 9–17) using the difference in the compositional dissimilarity between between-group pairs and within-group pairs. A t-test was employed to assess the differences in the resistance or resilience between fungal and bacterial communities at each time point in each compartment. To account for the differences in the taxonomic resolution of 16 S and ITS[57], we compared bacterial

16 S OTUs against both fungal, species-level ITS OTUs, as well as fungal families. We reported only results that are consistent across these two conditions.

To test $H_1$ and $H_2$ at all-correlation level, Spearman correlation (Rho, $\rho$) matrices between OTUs were calculated using the corr.test command in psych 2.2.5 package[58]. The correlation matrices were split into three subsets: i) correlation between bacterial and bacterial OTUs (BB), ii) correlations between fungal and fungal OTUs (FF), and iii) correlations between bacterial and fungal OTUs (BF). Differences in the distribution density of Spearman $\rho$ between control and stress treatments were visualized in ggplot2 version 3.3.6[59], and were evaluated by a linear mixed effect model that included random effects of compartment (leaf, root, rhizosphere and soil) and link types (BB, FF and BF) using the lme command in the lme4 version 1.1.29 package[60].

We analyzed networks for each period and treatment separately, following previous studies[61–64], to assure > 25 communities per network[65]. Thus, the drought-period network was based on 36 communities (6 plots * 6 time points) and the rewetting period network was based on 48 communities (6 plots * 8 time points). Concern about temporal autocorrelation, leading to spurious correlations among independent time-series, led us to use the approach of Coenen, et al.[66] to simulate 6 random walks (mimicking the drought period) and 8 random walks (mimicking the rewetting period) of 6 time series (mimicking our six replicating samples). We were unable to detect significant temporal autocorrelation among the 15 comparisons of six, random time series for either the drought period ($\leq$ 1 significant association, Supplementary Fig. 15a, b) or rewetting ($\leq$ 1–3 significant associations, Supplementary Fig. 15c, d). In each of these analyses, we only used taxa that occurred in at least 8 communities, following Shi et al[67] and de Vries et al.[19]. To test $H_1$ and $H_2$ at co-occurrence level, the above-mentioned Spearman correlations with $\rho > 0.6$ and $P < 0.05$ (the $P$ value was adjusted using FDR method) were retained (Supplementary Table 3). The taxon-taxon-space approach was employed to detect the proportion of significant taxon-taxon associations are likely driven by dispersal limitation[68]. In addition to FDR, we used Random Matrix Theory (RMT) to assess the robustness of correlations as implemented in the Molecular Ecological Network Analyses Pipeline (MENAP)[69]. We found that all empirical networks were non-random (Supplementary Table 4). We then compared the association networks based on Spearman correlations as filtered by either the FDR or RMT approaches, finding that results of these two different methods are consistent in terms of drought response (Supplementary Figs. 16, 17). Note that in only one case, roots, is there disagreement where the FF network showed disruption using the FDR approach but was unchanged using the RMT approach. We also constructed the network using the Pearson correlation and CoDa method[70] (Supplementary Figs. 18, 19). Co-occurrence networks were constructed using the graph.data.frame command in igraph package 1.3.1[71]. Modularity was defined as the measure of how much of the network is structured as cohesive subgroups of nodes (modules) in which the density of associations was higher within subgroups than among subgroups[45,72]. Network modules were detected using the cluster_fast_greedy command, and network modularity was calculated using the modularity command in igraph 1.3.1 package. Network hubs were detected by calculating $Pi$ using the part_coeff command and $Zi$ value using the within_module_deg_z_score command in igraph package. Finally, an AMF-related network was constructed by retaining significant positive co-occurrences between AMFs and other fungi, and between AMFs and bacteria.

**Reporting summary**. Further information on research design is available in the Nature Research Reporting Summary linked to this article.

## Data availability

No new data were generated in this study. The data used in this study had been deposited in the Sequence Read Archive database under accession code PRJNA412410, PRJNA494573, PRJNA435634, PRJNA435642, and PRJNA435643. Source data are provided as a Source Data file. Source data are provided with this paper.

## Code availability

All scripts used in this study are available at GitHub (https://github.com/ChengGaoBerkeley/EPICON.FunBac) archived on Zenodo: https://doi.org/10.5281/zenodo.6614679.

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

## Acknowledgements

The research was supported by the program of Systems Biology Research to Advance Sustainable Bioenergy Crop Development of DOE BER (DE-FOA-0001207), under the fund #DE-SC0014081. The work conducted by the US DOE Joint Genome Institute is supported by the Office of Science of the US Department of Energy under Contract no. DE-AC02-05CH11231. CG is also financially supported by the National Natural Science Foundation of China (32170129), Strategic Priority Research Program of the Chinese Academy of Sciences (XDA28030401), and State Key Laboratory of Mycology (SKLMZD202101).

## Author contributions

C.G. and J.W.T. conceived of and wrote the manuscript for the fungal and bacterial portion of a larger project conceived of by P.G.L., D.C.D., E.P., J.V., J.A.D. and J.W.T. with P.G.L. as director. P.G.L., R.B.H., J.H., J.A.D. and M.M. coordinated the field work and sample storage. C.G., L.M. and L.X. performed the molecular analysis. C.G. performed the bioinformatics and statistical analyses, and advised by V.S., L.C., E.P. and D.C.D.

## Competing interests

The authors declare no competing interests.
