## [Peer Review File · Nature Communications]

Reviewer comments, first round –

Reviewer #1 (Remarks to the Author):

This study investigated the resilience and resistance of Sorghum-associated bacterial and fungal communities against drought. The strength of the study for me is that it targeted both bacteria and fungi and studies these communities in all relevant soil and plant compartments (soil, rhizosphere and leaf). The authors rightly point out that most studies have focused on bacteria alone, and often on a single microbial compartment.

I found the study interesting and believe it will interest others in the field, as there is considerable interest in understanding resilience and resistance in microbial communities, and this study's comprehensive experimental design makes it a likely important article for those in the field.

The authors analysed and discussed positive microbial interactions and related this to resilience and resistance. As many of the studies that linked positive interactions with resilience and resistance are based on macro-ecological studies, it would be relevant to provide some context for microbial studies which also looked more specifically at positive associations in networks. There are a few studies that looked at the ratio of positive interactions in microbial networks in relation to ecological status, especially ecological succession, and the authors didn't mention these studies. I suggest including some of these studies in their discussion: e.g. 10.3389/fmicb.2019.02887; 10.1038/ismej.2014.54; 10.1111/1751-7915.13487; 10.3389/fmicb.2015.01200 and references therein.

On a more general note, the authors take the correlations as an indication of interactions, however, correlations may derive from habitat-filtering or dispersal limitation processes, in which case, inferring resilience and resistance may be less straightforward. Indeed, this may be a possible reason for some of the results obtained by the authors which led to the rejection of some of their hypotheses. In other words, if the authors were able to remove correlations that were not due to habitat filtering, and particularly due to dispersal limitation, then the remaining correlations may support their original hypotheses. Partitioning correlations due to dispersal limitation, habitat filtering and interactions may not be possible in most cases, but for soil samples (and perhaps rhizosphere), it may be possible if the authors have the spatial coordinates for each sample (more details here: 10.1111/1755-0998.13079). At least the authors should acknowledge the issue that correlations may be due to other processes than interactions.

In lines 176-179 the authors state that resistance is $1-R_2$ (when comparing control and droughted communities) and resilience is $1-R$ (when comparing control and re-wetted communities). However, in lines 180-186 (and in the figures and tables), the authors detail and discuss R_2 values, rather than $1-R_2$. Perhaps this can be simplified? Since R_2 is related to the level of change between treatments, perhaps the $1-R_2$ definition is not needed?

In Figure 1 is the significance indicated in every compartment for what comparison exactly? Could the authors detail this in the legend? As the authors use the R_2 as a measure of resilience and resistance, to claim for instance that "that the fungal mycobiome is more resistant than the bacterial microbiome to both pre- and post- flowering drought", it would be important to show that these differences in R_2 between bacteria and fungi are significant.

Other comments:

Line 248 "rewatered" should be re-wetted for consistency.

Line 258: Is network modularity determined by the number of modules detected?

Line 341, what is -- for?

Lines 467-469: were the p-values corrected for multiple testing? If not, why?

Reviewer #2 (Remarks to the Author):

The ms describes a new analysis based on the recombination of two previously published datasets that examines resistance and resilience of microbial communities (bacteria and fungi) associated with leaves, roots, rhizosphere, and surrounding soil of an agricultural crop, sorghum, subjected to

drought stress. Using modern methodology (e.g., rDNA metabarcoding) the group finds that drought disrupts the plant-associated microbial communities and that co-occurrence networks among functional guilds of rhizosphere fungi and leaf bacteria were "dramatically strengthened" in the pre-flowering drought treatment. The ms frames these findings within the context of the classical stress gradient hypothesis, and also suggests that microbial 'hub' taxa could be identified that might have utility as seed-taxa serving to support the microbial communities overall under drought stress expected with climate change scenarios.

While I feel that the ms represents an important contribution to the field, especially considering the cited deficits of previously published works that focus on agriculture, I feel that the ms is not yet ready for publication. I recommend a significant revision that addresses the following important issues:

>>Language awkwardness/precision/directness: I find the language of the ms to be very awkward in several sections and in some areas the language also lacks precision and also could be edited to be more straightforward. I present a few examples (primarily from the Introduction) here...

Introduction: starting @ line 57 (drought...drought/plant gene...plant genes) - "When drought curtails photosynthesis in response to drought the most profound change in plant gene transcription is the down regulation of plant genes involved in managing microbial association and this change in expression correlates with a decline in the abundances of these root-associated microbes." ... consider, "One of the most profound changes in plant transcription in response to drought is the down regulation of genes involved in managing microbial association that can result in a reduction in abundance of root-associated microbes."

Introduction: starting @ line 81 - "We surveyed previous research that included both fungi and bacteria from the perspective of the community compositional response to drought and subsequent rewetting (Table S1) finding that H1 has been both supported and falsified, and H2 has been either falsified or untested." ... consider, "We surveyed the literature for research that addressed community composition shifts, for both fungi and bacteria, in response to drought and subsequent rewetting (Table S1) finding that H1 has been both supported and refuted, while H2 has either been refuted or remains untested."

Introduction: starting @ line 96 - "Here, to advance our aim of including microbe-plant interaction in efforts to combat crop loss due to drought, we test these hypotheses, H1 and H2, through comparisons of microbial communities in four compartments (leaf, root, rhizosphere and soil) in fields of sorghum during these three treatments, when drought imposed prior to flowering, when this preflowering drought is relieved by watering, and when drought is imposed after flowering." ... from my reading of the methods, this study was carried out during drought conditions (i.e., it was not "imposed") in CA with crops being subjected to watering (rewetting) or not, consider, "In this study we focused on both bacterial- and fungal-plant interaction, examining hypotheses H1 and H2 for microbial communities associated with sorghum leaves, roots, rhizosphere, and surrounding soils in agricultural fields under drought conditions that were relieve post-flowering by watering or not." Further, while I agree that the results from this study provide insights that might be helpful in efforts to "combat crop loss", the study did not directly address "crop loss" and, therefore, statements such as this are likely a bit of an overreach.

Introduction: starting @ line 109 (Bacteria are typically considered a Domain, while Fungi are typically considered a Kingdom) - "For example, regarding drought stress, it has been proposed that positive interactions should increase in frequency under stressful conditions, a response explained by the stress gradient hypothesis. It also has been proposed from studies of microbes on Arabidopsis leaves, roots and soil, that correlations between microbes within kingdoms tend to be positive, while correlations between kingdoms tend to be negative. Additionally, ecological modeling has indicated that negative interactions should promote stability...." ... consider, "For example, it has been proposed that positive microbial interactions should increase in frequency under stress scenarios, such as drought, a response explained by the stress gradient hypothesis (SGH). Further, stress studies of microbes on Arabidopsis leaves, roots, and the surrounding soils suggest that within-taxonomic group microbial interactions tend to be positive, while those between-taxonomic groups are negative. Ecological modeling also indicates...." Further, microbial

interactions, which biological/ecological in nature, should not be confused with correlation, which is simply a statistical method. For example, positive correlations related to shifting microbial abundances might be interpreted as mutualist interactions (or facilitation), while negative correlations might be interpreted as antagonistic interactions (or competition). The paper tends to confuse these concepts a bit (see comments immediately above and below), and the authors should bear in mind that they are attempting to view/interpret microbial interactions through the lens of statistical correlation (e.g., correlations metrics are appropriate for the results, but the interpretation (i.e., in discussion) should focus on the interactions).

Introduction: starting @ line 112 - "Using these studies to frame hypotheses at the all-correlation level, for our resistance hypothesis, H1, under drought we expect an increase in the proportion of positive correlation most strongly for B-B, followed by F-F, and lastly by B-F correlation; and for our resilience hypothesis, H2, under re-watering, we expect a decrease in the proportion of positive correlation most strongly for B-B, followed by F-F, and lastly by B-F correlation." This sentence does not entirely make sense given the discussion as the proposed hypotheses are not: A) clearly defined overall; B) completely consistent with the studies mentioned; or C) differently defined for resistance vs. resilience - also, I'm not sure what phrases like "all-correlation level" mean ...consider, "These previous studies provide a framework for the hypotheses we propose here, namely under the stress of drought, we expect enhanced facilitation within taxonomic groups (i.e., positive correlations for B-B and F-F) and enhanced competition between taxonomic groups (i.e., negative correlation for B-F). Further, the hypotheses proposed by authors in the ms need to be distinguished from those of other work (i.e., those associated with SGH) and more clearly defined and consistent overall. For example, the hypotheses mentioned in the abstract focus on fungi and state that fungi are "(i) more resistant but (ii) less resilient than bacteria" (we assume this refers there respective status under the stress or drought), while the H1 and H2 mentioned here focus on interactions.

Introduction: paragraph @ line 118-136 - I find this paragraph to be confusing and repetitive with respect to the hypotheses (and see above) overall, the discussion of "nonintuitive outcomes" is a bit obtuse and appears to be splitting hairs (to justify results/methods?). Also, "Simplifying matters by focusing on just the significant, positive correlations" - if a correlation is not significant then it should not be considered as a result at all; further, the paragraph above and H1/H2 stress the importance of validating negative correlations. This paragraph appears to be justification for the methods used in the co-occurrence network analysis part of the study, but the case could be more clearly and directly made (i.e., this is a common method for such analyses).

Introduction: starting @ line 137 - The authors should note different terminology typically use in distinguishing between network element vs. network properties. For example, 'modules' are network elements (functional units of connectedness within the network) whereas 'modularity' is a network property (the characteristic of being divided into multiple modules); likewise, 'hubs' are network elements (nodes with a number of links/edges that greatly exceeds the average) and 'hub emergence' (networks that reflect the characteristic of contain multiple highly linked hubs).

Introduction: starting @ line 148 - "Our experimental system is an agricultural field....Compared to previous studies, our system is simpler because it has just one plant genotype, which is grown in synchrony...Our identification of bacteria and fungi by DNA sequence is more precise...." Etc.... rather than directly comparing the work carried out here to previous studies, it might be more preferable to simply state the strengths of this study (the relative improvement over earlier work should already be clear from justifications provided in previous chapters within the Introduction), consider "Here we use modern high-throughput sequencing techniques to examine interactions of microbial communities, bacterial and fungal, associated with leaves, roots, rhizosphere, and surrounding soils of two sorghum cultivars planted as a monocultures in agricultural fields during a period of drought. This experimental system allowed us to investigate resistance and resilience of these microbial communities under the stress of drought and subsequent recovery after watering...etc."

>>Questions related to approach, interpretation, and statistics used:

Ecological concepts: The authors state, "We use definitions of ecological resistance as the change

in compositional dissimilarity in response to stress and of ecological resilience as the recovery in compositional dissimilarity when stress is relieved. Ecological resistance and resilience are determined by comparing compositional dissimilarity among communities within treatments (combined control and stress) with dissimilarity between control and stress communities. Specifically, resistance is $1-R_2$ using control and droughted communities and resilience is $1-R_2$ using control and rewetted communities, in which R_2 was determined by permutational analysis of variance (permanova 40)." The authors should directly cite works influencing the definitions here, for example the referenced paper Shade et al. 2012 provides excellent discussion over the concepts of resistance and resilience as well as related terminology. These authors state, "Disturbance and community stability are necessarily related, as stability is defined as a community's response to disturbance (Rykiel, 1985). Here, we adopt definitions most similar to Pimm (1984), in which stability is comprised of resistance and resilience (Table 1), two quantifiable metrics that are useful for comparing community disturbance responses and have precedent in the microbial ecology literature (e.g., Allison and Martiny, 2008). ... Here, resistance is defined as the degree to which a community is insensitive to a disturbance, and resilience is the rate at which a community returns to a pre-disturbance condition (Pimm, 1984)." These authors further define the related 'Stable state' as, "A condition where a community returns to its original composition or function following disturbance." As the ms authors base their analyses on Bray-Curtis dissimilarity, it should be noted that here a value of 0 means two sites that have the same community composition (they share the same species at the same levels of abundance), whereas a value of 1 denotes two communities that are completely dissimilar (i.e., they do not have species in common). Given this, could resilience, for example, be better defined as "the recovery in compositional similarity (i.e., Bray-Curtis dissimilarity values converging on zero)." Such a definition would have bearing, for example, on the interpretation of Figure 1. Further, this figure also stresses the reliance on the R-squared value (inversely proportional to the effect strength) in interpreting resistance or resilience, yet the generally low R_2 here suggests very little variation in distances is explained by the groupings - are we to believe that this means (inversely) very strong resistance or resilience effects? Further, p-values in Permanova type are strongly influenced by sample size, was this accounted for in the analysis (similarly see comments regarding FDR below). Some of these issues need to be cleared up.

Network analysis: When running numerous parallel correlations, as are possible with metabarcoding sequence data, the chance of recovering spurious significant positive correlations are greatly enhanced. There are statistical methods, such as FDR (false discovery rate), that can be used to reduce the influence of false positives. This may be especially true for non-parametric approaches (i.e., Spearman's ranked correlation). Corrective measures (i.e., FDR), may be warranted here to reduce type I errors.

Guild approach: The authors also use a fungal guild concept in their network analyses, while these concepts appear to be derived from the paper below, yet the authors do not directly cite this paper/source/software and should (especially in the methods):

Nguyen NH, Song Z, Bates ST, Branco S, Tedersoo L, Menke J, Schilling JS, Kennedy PG. 2016. FUNGuild: an open annotation tool for parsing fungal community datasets by ecological guild. *Fungal Ecology* 20:241-248.

Further, care should also be taken when interpreting the network analysis results. For example, the authors claim that "co-occurrence networks among functional guilds of rhizosphere fungi ... were dramatically strengthened by pre-flowering drought", yet Figure 2B shows that the "strengthened" network contains numerous saprotrophs and plant pathogens, suggesting that "pre-flowering drought" contributed to decay (perhaps of dead plant matter) and disease.

Also see comments above regarding potential overreaching statements.

Examples of other issues:

Introduction: starting @ line 153 - "seedling emergence to fruit maturation" ... as sorghum is a member of the Poaceae (i.e., a grass) the seed (e.g., millet) of sorghum is typically referred to as a cereal grain rather than a "fruit".

Results: starting @ line 165 - "As noted above, the simple fact that fungi grow more slowly than bacteria...." I don't feel that this is a simple matter, bacteria "grow" as single-celled microorganisms through binary fission where, yes, doubling times can range in 10s of mins. Growth for fungi is something completely different; a (sometimes massive) multicellular (generally) mass of hyphae (a mycelium) that grows by extension at the hyphal tip (unless we are talking about yeasts), where some taxa (e.g., *Neurospora*) can have relatively high growth rates (e.g., several mm per hour) at the hyphal tip. Therefore, a reductionist approach to growth rates is likely not warranted here.

Results: paragraph @ lines 165-179 - There are no results given here, this paragraphs has elements that may be more appropriate for the Methods section.

Reviewer #3 (Remarks to the Author):

In this manuscript, the authors report the effect of pre-flowering drought, post-flowering drought, and recovery after pre-flowering drought on fungal and bacterial communities and networks in/on roots, rhizosphere soil, bulk soil, and leaves of field-grown sorghum. They hypothesise, based on previous work, that fungal communities and network are more resistant but less resilient than those of bacteria. They test these hypotheses using previously published data for new analyses. They find that their hypothesis that fungal communities are more resistant and less resilient than bacterial communities is supported. Using all correlations between bacteria and fungi in the four compartments, they find that the frequency of positive correlations increased in pre-flowering drought, but using only significant positive correlations (ie co-occurrence networks), they find that pre-flowering drought disrupts networks in roots, rhizosphere and soil but increases their connectivity on leaves. Re-watering resulted in networks resembling control networks again, except for the network in soil (but note that I inferred those results myself from Fig. 3 as I found the description of the results hard to follow). They conclude that understanding microbial network response to stress might inform manipulating microbial communities for increased plant tolerance to stress in agricultural settings.

I enjoyed reading this mostly clearly written manuscript that addresses interesting hypotheses. However, I found the amount of results presented quite overwhelming and not always easy to follow/ interpret. The hypotheses stated are quite abstract and informed entirely by previous work on soil fungal and bacterial communities and network responses to drought, and in that sense the paper reads as largely confirmatory and leans heavily on the results from a few recent papers. I also feel that there is really a severe lack of context on why we want to understand how the communities/ networks in these different plant compartments respond to drought. To me, it would be much more interesting to focus in on the differences between these compartments. What drives the assembly of fungal and bacterial communities on leaves, and how is this different from those in roots and in soil? What would be the implications for their functioning and for plant health of the changes in these communities in response to drought? I am missing all of this in the manuscript, other than quite vague and general statements. I would suggest to focus on this, and I would also suggest ditching the post-flowering drought treatment, as there is no recovery phase after this drought, which makes it difficult to compare these data to the pre-flowering drought.

Moreover, while the manuscript focusses on networks, never is the reliability of these correlations and whether they actually represent interactions between microbes discussed. Positive correlations between microbes can simply indicate niche sharing or responding to the same drivers. Moreover, it is not clear which OTUs were used for correlations (all? Or the ones that occurred over a certain number of experimental units? Or the most abundant ones?), and on how many observations these correlations are based. From the methods it seems that there were 6 replicates of each treatment – does this mean that correlations were based on only 6 data points? Then I would seriously question the robustness of the resulting networks.

In addition, while on close inspection the analyses seem robust and the results are mostly correctly interpreted, I found the figures quite hard to understand as the axes and legends are

rather ambiguous. The clarity can be improved, and perhaps also the presentation, because as I said above the amount of data is overwhelming.

More detailed comments:

L 164: yes, but also because of their hyphal growth form and thick cell walls, see Schimel et al. 2007 Ecology and Guhr et al. 2015 PNAS.

L 175-184 and Figure 1: I found this section very hard to follow. Here, it says that resistance and resilience are calculated as $1-R_2$, but in the figure Bray-Curtis dissimilarities are reported (are similarities? This is not clear), and in the figure legend it says resistance and resilience. I am lost. It's also not immediately clear what is meant by inter-group and intra-group.

L 205: can you be more specific? Which compartments?

L 238-244: I found this section very hard to read, as pretty much every sentence mentions that vertices are dropped and rise, but in response to what and compared to what? I assume to drought, but this is never explicitly mentioned.

L 252: The biotic interactions become even more complex than the control after rewatering. But is this resilience? Resilience means that the disturbed treatment is approaching or resembling the control.

L 315-318: I don't understand this sentence

L 325: not just in leaf in post-flowering drought, also in soil and root

L 324: De Vries et al. 2018 Nat Comms also analysed combined bacterial-fungal networks – this is detailed in their supplementary material

L 327-330: this sentence makes no sense to me. Hypotheses developed from one type of analysis? I would think that it is not about the analysis but about the concept. The analysis is just a means to test a hypothesis.

L330-331: again, I have no idea what is meant here. Whole communities hide variation based on compartments?

L332-334: I think it is rather stark to make inferences about applications in agriculture from these theoretical hypotheses

Methods: I understand that these are previously published data but there's really more detail needed here. How large were the plots? What was the experimental layout? How were samples collected? What other analyses were done? Were there six replicates per treatment, and does this mean that correlations for network analyses were done only using 6 datapoints....?

Reviewer #4 (Remarks to the Author):

Cheng Gao and colleagues in their manuscript 'Resistance and Resilience in Microbes: Co-occurrence Networks Delve Deeper Than Community Composition' address two fundamental questions in the field of microbial community compositions: Resistance and resilience. To do so, they combine two very comprehensive previously published datasets analysing microbial communities on crop plants under extreme drought conditions and irrigation. The datasets are based on 16S and ITS amplicon sequencing and the analyses in the paper is primarily based on pairwise correlations of these datasets.

Particularly the question if fungi are more resistant H1 but less resilient H2 than bacteria is certainly a key question in the field and addressed in depth in this manuscript. Besides direct analyses of correlation data, the authors use networks to get deeper insights into community structures. They identify a disruption of communities by drought and see an increase of positive correlations among bacteria, fungi and across kingdoms correlating bacteria and fungi. In combination with network analyses, this gives support for the stress gradient hypothesis. Based on their analyses, they can further underpin the importance of mycorrhiza fungi in stabilizing communities under drought.

In summary, the paper touches a very timely and relevant field and the authors show convincingly that their dataset can be used to infer their central hypothesis H1 and H2. Although I think this manuscript has great potential it would certainly benefit from more details and by addressing some of the following points:

1. As the authors state, key to the paper are pairwise correlations. The authors focus, however, only on Spearman's Rho or Spearman's rank-order correlation. This assumes a monotonic relationship. From the paper it is not clear if the authors have analyzed other correlations to show that this fits the best or have plotted the data to see if this really fits for all samples. Why not using Spearman's correlation, particularly for the networks this might be a better choice or a combination?
2. Further to the correlation analyses: How valid is it to correlate 16S and ITS data together to make conclusions about robustness and resilience? Both will result in completely different resolution. ITS is used to resolve on a species level, 16S will rarely branch that deep. Wouldn't it be better to compare 16S and 18S? Is it possible that bacteria are more resilient because of less resolution, meaning other bacteria move in following rewetting but they are seen as having the same 16S sequence while fungi move back in that show the same taxonomic distance but can be resolved?
3. Very much depends on the calculation of the networks. From the methods I can see igraph has been used and the implemented calculation of networks. To better understand the quality and robustness of the networks it certainly needs more information on the calculation. For example, how was sparsity addressed and how density of the networks. Based on the figures, density is a particular issue, as very dense networks are compared to extremely sparse networks. I would suggest to use at least one other method to calculate the networks correcting for abundance and sparsity or not correcting and comparing those to each other. In my opinion this is relevant to identify if modularity is robust, as this has been debated a lot.
4. As far as I understand from the data sets, the samples are not independent from each other but have a time factor: PRE-Drought, PRE-Rewatering, POST-Drought. To analyze stability it would be useful to track vertices over time and compare PRE and POST networks directly. Particularly positional stability of each vertex would be a good additional measure when comparing different network calculations.
5. A minor thing but relevant to understand what has been done: What are the Guilds and how have they been calculated? I guess this is based on Nguyen et al 2016 but I could not find any information.
6. Question concerning the experimental layout: The experiments have been set up in an area with extremely low precipitation. So any microbe in the soil would be adapted to cope with drought. In this case I would assume that regular irrigation is a perturbation to the community and not drought. Have samples been taken before the planting that could be compared? Is the drought state perhaps a communal 'recovery'?

**Response to reviews.**

**Resistance and Resilience in Microbes: Co-occurrence Networks Delve Deeper Than**
**Community Composition**

Cheng Gao^{1,2,9*}, Ling Xu^{2,3,9}, Liliam Montoya², Mary Madera², Joy Hollingsworth⁴, Liang Chen⁵,
Elizabeth Purdom⁶, Vasanth Singan⁷, John Vogel^{2,7}, Robert B. Huttmacher⁸, Jeffery A. Dahlberg⁴,
Devin Coleman-Derr^{2,3}, Peggy G. Lemaux², John W. Taylor^{2*}

We begin with responses to seven general concerns and then move to the comments of
individual reviewers. To reduce redundancy, where individual reviewers reiterated the general
concern, we refer the reader back to our response to the general concern.

**GENERAL CONCERN 1** - Alternative analyses to account for different taxonomic resolution of
16S and ITS data (Reviewer 4).

**Response:** We, too, share reviewer #4's concern that 16S and ITS identify bacteria and fungi at
different levels of taxonomic resolution (Bruns & Taylor 2016 Science). Ideally, we would deepen
the level of taxonomic resolution for bacteria to the species level. However, we (and all other
researchers) are limited at the present to 16S rRNA (about the family level or higher) for
characterizing bacterial communities.

To address the reviewer's concern about the different resolution of 16S and ITS, we compared
bacterial 16S OTUs against both fungal communities recognized by ITS OTUs as well as fungal
communities recognized at the family level (roughly the taxonomic level determined by 18S
rDNA). The results of analyses using either fungal families or OTUs are consistent. Out of total 36
comparisons (15 root, 15 rhizosphere and 6 soil), different family and OTUs results were detected
in four instances. In two of these, significances detected by OTUs were not detected by family
(root, week 4 and 17) and, in the other two cases, significances detected by family were not
detected by OTUs (rhizosphere, weeks 7 and 8). In our revised manuscript we report only results
that are consistent in both analyses. Importantly, our key findings that fungi are (i) more resistant
than bacteria to drought stress but (ii) less resilient than bacteria when the stress is relieved by
rewetting are unaffected by this change because of the 23 significant comparisons supported by
both analyses from weeks 5 and 9-16 in root and weeks 4-6 and 11-17 in rhizosphere.

**Revised figure:**

**Fig. 1. Resistance and resilience of bacterial and fungal community composition.**

Ecological resistance to drought stress is detected by comparing compositional dissimilarity of
between-group pairs (control-drought pairs) against within-group pairs (control-control pairs and
drought-drought pairs) at each of the droughted weeks (weeks 3 - 8). Ecological resilience to
rewetting is detected by assessing, from before to after rewetting, the change in the difference
of compositional dissimilarity between within-group pairs and between-group pairs. Here, the
point just before rewetting was week 8 and the points after rewetting were weeks 9 - 17. To
account for the different resolution of ITS and 16S, we compared bacterial 16S OTUs against both
fungal ITS OTUs as well as fungal families. In 32 of 36 cases, the results of fungal families and
OTUs are consistent. Different family and OTUs results were detected in two cases where
significances detected by OTUs were not detected by family (root, week 4 and 17), and in two
cases where significances detected by family were not detected by OTUs (rhizosphere, weeks 7
and 8). We report only results that are robust across these two conditions. Significantly higher
resistance to drought of fungi than bacteria was detected in root (week 5), rhizosphere (weeks 4
52 - 6) and soil (weeks 4, 6 - 8). Significantly higher resilience to rewetting of bacteria than fungi
was detected in root (weeks 9 - 16) and rhizosphere (weeks 11 - 17). Note that fungi exhibited
stronger resilience than bacteria at the first week of rewetting (week 9). The finding that fungal
community composition in soil is not shaped by drought prevented us from further detecting
resilience in this compartment. Note that fungal communities in early leaves are excluded from
analysis due to the high proportion of non-fungal sequencing reads. The detailed results at fungal
family levels can be found in Fig. S1.

GENERAL CONCERN 2 – additional network analyses (Reviewers 2 and 4),

Response: In addition to the Spearman method of network analysis used in our original
manuscript, we added network analyses using the Pearson correlation method and the CoDa
method of Gloor et al. 2017. In almost all cases, the results of these three different methods are
consistent.

We present the results of the Spearman analysis in the manuscript (Figs. 2, S3), because the
Spearman method is widely used in ecological research (e.g., de Vries et al 2018 Nat Commun).
We also present the results using the Pearson and CoDa approaches as supplementary
information (Fig. S14, S15).

Our first conclusion, that drought in general disrupts microbial networks, was found in 11 of 13
Spearman networks, 10 of 13 CoDa networks, and 9 of 13 of Pearson networks. The two out of
13 cases where the Spearman result was not supported by other methods are: i) The BF network
in rhizosphere was judged to be disrupted by drought using the Spearman and CoDa methods
but was found to be enhanced by drought using the Pearson method; and ii) The FF network in
soil was judged to be disrupted by drought using the Spearman method but was judged to be
unchanged by the Pearson and CoDa methods.

Neither did these new analyses have any effect our second conclusion, that co-occurrence
networks among functional guilds of rhizosphere fungi and leaf bacteria were dramatically
strengthened by drought, because these same results are found in all the three methods.

Importantly, all three methods support the key findings that: (i) *In general, drought disrupts*
*microbial networks based on significant positive correlations among bacteria, among fungi and*
*between bacteria and fungi. (ii) In contrast, co-occurrence networks among functional guilds of*
*rhizosphere fungi and leaf bacteria were dramatically strengthened by drought.*

Revised and added figures:

 **Fig. 3 Networks of significant positive cross-taxonomic group correlations (bacteria and fungi).**
 (A) Fungal OTUs (blue) and bacterial OTUs (black) are graphed as nodes. Significant positive
 Spearman correlations are graphed as edges ($Rho > 0.6$, $FDR P < 0.05$); Skyblue (fungus-fungus),
 grey (bacterium-bacterium) and red (bacterium-fungus). All three types of co-occurrences (BB,
 FF and BF) are generally disrupted by drought (but not FF in rhizosphere and BB in leaf, see Fig
 S3), and recovered by rewetting. (B-C) FF co-occurrences in rhizosphere and BB co-occurrences
 in leaf are drastically enhanced by drought, which is coupled with the increase of the proportion
 of interaction between fungal guilds and the increase of the proportion of interaction between
 bacterial phyla. The key finding that drought enhanced the rhizosphere fungal network and the
 leaf bacterial network was also supported by the Pearson method and CoDa methods (Figures
 S14 and S15).

 **Fig. S3 Subnetworks of significant positive Spearman correlations (A) between fungal taxa and**
 **(B) between bacterial taxa. (A)** Subnetworks of significant positive correlations between fungal
 OTUs. The FF co-occurrence in rhizosphere is drastically enhanced by drought, although it is
 strongly disrupted in root. Re-watering caused recovery of the FF network, with
 overcompensation in root and a lag in rhizosphere and soil. (B) **Subnetworks of significant**
 **correlations between bacterial OTUs.** The BB co-occurrence in leaf is drastically enhanced by
 drought, although it is strongly disrupted in root, rhizosphere and soil. Re-watering caused
 recovery of the BB network.

 **Fig. S14 Co-occurrence network using the Pearson method. (A)** The fungal co-occurrence
 network in rhizosphere is drastically enhanced by drought, although it is strongly disrupted in
 root. (B) The bacterial co-occurrence network in leaf is drastically enhanced by drought, although

it is strongly disrupted in root, rhizosphere and soil. Rewetting caused recovery of both fungal and bacterial networks.

 **Fig. S15 Co-occurrence network using the CoDa method. (A) The fungal co-occurrence network**
 **in rhizosphere is enhanced by drought, although it is strongly disrupted in root. (B) The bacterial**
 **co-occurrence network in leaf is drastically enhanced by drought, although it is strongly disrupted**
 **in root, rhizosphere and soil. Rewetting caused recovery of both fungal and bacterial networks.**

GENERAL CONCERN 3 - correcting p-values for multiple comparisons if not done so
 (Reviewers 1 and 2),

**Response: We now correct p-values using the FDR method. Our key findings were not changed**
 **by the FDR correction of p values. We have provided this information in our revised ms.**

**Revised: To test H_1 and H_2 at the co-occurrence level, the above-mentioned Spearman**
 **correlations were retained where $Rho > 0.6$ and $P < 0.05$; the P value having been adjusted using**
 **the FDR method.**

GENERAL CONCERN 4 - providing more information on the methods (all Reviewers, e.g. see
 Reviewer 3's comment on sample size)

**Response: We added information about methods and clarified the number of samples in the**
 **analysis.**

**Added text: We analyzed networks for each period and treatment separately. Thus, the drought-**
 **period network was based on 36 communities (6 plots * 6 time points) and the rewetting period**

network was based on 48 communities (6 plots * 8 time points). In each of these analyses, we
 only used taxa that occurred in at least 8 communities, following Shi et al (Shi et al. 2016) and de
 Vries et al (de Vries et al. 2018).

GENERAL CONCERN 5 - . . . and toning down the interpretation of co-occurrence as evidence
 of interactions (all Reviewers).

**Response: We have toned-down the interpretation of co-occurrence as evidence of interactions.**

Added text: We note that correlation does not necessarily equate with interaction, but also can
 be ascribed to habitat-filtering, niche sharing or dispersal limitation (Goberna et al. 2019). As is
 the case with most field-based experimental designs, it is not possible to assess the effect of
 habitat filtering and niche sharing. However, we can note that the role of dispersal limitation on
 the co-occurrence network is weak. Based on our implementation of a taxon-taxon-space
 association approach, the percentage of network links related to spatial distance was no more
 than three percent (0 – 2.94 %; Figure S13). This result echos the absence of a significant
 relationship between spatial distance and dissimilarity of microbial community composition
 reported in our previous study (Gao et al. 2020). Thus, dispersal limitation is not likely the driver
 of microbial interaction and community composition in our small research site (~480m²), which
 has been cultivated for nearly six decades and was planted to one crop (sorghum) throughout
 our study (Gao et al. 2020).

**Fig. S13 Proportion of taxon-taxon associations related to dispersal limitation.** For each of taxon
 (A) -taxon (B) pair in the co-occurrence network, dispersal limitation was regarded as the driver
 if both taxa showed significant correlation with spatial distance.

Please refer to the full reports below for details. Without substantial revisions, we will be unlikely
to send the paper back to review.

Additionally, another reviewer who did not provide a full report raised a potential concern on the
public sorghum drought data that may have been included in the analysis, namely low quality
scores of some of the deposited sorghum data. This point should also be addressed.

GENERAL CONCERN 6 - low quality scores of some of the deposited sorghum data.

Response: We have found a high proportion of non-specific amplification in fungal data of early
leaf samples. We removed these data when making this revision of the manuscript. Because none
of the results concerning these data are key findings of our report, we no longer report that: i)
early leaf fungal community composition was not affected by pre-flowering drought; ii) early leaf
fungal correlations was not affected by drought; and iii) early leaf fungal network was not
changed by drought.

Added text: The proportion of fungal reads was low in early leaves (weeks 1- 8) due to non-
specific amplification (Gao et al. 2020), so we excluded these fungal data from our analyses.

If you feel that you are able to comprehensively address the reviewers' concerns, please provide
a point-by-point response to these comments along with your revision. Please show all changes
in the manuscript text file with track changes or colour highlighting. If you are unable to address
specific reviewer requests or find any points invalid, please explain why in the point-by-point
response.

GENERAL CONCERN 7 – Determination of resistance (1-R²) and resilience (as 1-R) from
community composition (although this concern was not among those listed by the editor, it was
raised by more than one reviewer).

Response: Reviewer's note confusion caused by our usage of 1-R². Now we directly calculate
resistance and resilience following the methods of Shade et al 2012 and have removed the text
about 1-R² throughout our manuscript. As a result of this change, we now use a t-test to assess
significance in the differences in resistance and resilience between fungal and bacterial
communities. (note that revised Figure 1 addresses both General Concerns 1 and 7).

Importantly, our key findings are unaffected by this new analysis. As before, fungi being more
resistant to drought stress was supported at week 5 in root, weeks 4-6 in rhizosphere and weeks
4, 6-8 in soil, while fungi being less resilient than bacteria when drought stress is relieved by
rewetting was supported at weeks 9-16 in root and weeks 11-17 in rhizosphere.

Added text: We followed the approach of Shade et al. (Shade et al. 2012) to detect resistance
and resilience, which had been developed for univariate variables, e.g., richness. For multivariate
data, e.g., community composition, we modified it by calculating pairwise community

[revised manuscript text omitted]

We feel that we have addressed all the reviewer's general concerns. Below, we provide our
responses to specific comments and have incorporated our responses into our revised
manuscript, using track changes.

**Reviewer #1 (Remarks to the Author):**

This study investigated the resilience and resistance of Sorghum-associated bacterial and fungal
communities against drought. The strength of the study for me is that it targeted both bacteria
and fungi and studies these communities in all relevant soil and plant compartments (soil,
rhizosphere and leaf). The authors rightly point out that most studies have focused on bacteria
alone, and often on a single microbial compartment.

I found the study interesting and believe it will interest others in the field, as there is considerable
interest in understanding resilience and resistance in microbial communities, and this study's
comprehensive experimental design makes it a likely important article for those in the field.

The authors analysed and discussed positive microbial interactions and related this to resilience
and resistance. As many of the studies that linked positive interactions with resilience and
resistance are based on macro-ecological studies, it would be relevant to provide some context
for microbial studies which also looked more specifically at positive associations in networks.
There are a few studies that looked at the ratio of positive interactions in microbial networks in
relation to ecological status, especially ecological succession, and the authors didn't mention
these studies. I suggest including some of these studies in their discussion: e.g.
10.3389/fmicb.2019.02887; 10.1038/ismej.2014.54; 10.1111/1751-7915.13487;
10.3389/fmicb.2015.01200 and references therein.

*Original text:* We make use of all of these correlations to again examine H_1 and H_2 following the
lead of several previous studies. For example, regarding drought stress, it has been proposed that
positive interactions should increase in frequency under stressful conditions, a response explained
by the stress gradient hypothesis (Bertness and Callaway 1994, Callaway et al. 2002, Hoek et al.
2016, Velez et al. 2018, Hammarlund and Harcombe 2019, Piccardi et al. 2019).

**Response:** We appreciate the reviewer's suggestion to add information about publications on the
ratio of positive associations in microbiomes. We added relevant information from the
references provided by the reviewer, as well as the citations.

**Revised:** We make use of all of these correlations to again examine H_1 and H_2 following the lead
of several previous studies. Previous studies demonstrated that the percentage of positive
correlations is related to ecological factors that include succession, fertilization, and habitat (Dini-
Andreote et al. 2014, Faust et al. 2015, Sun et al. 2017, Jiang et al. 2018, Farrer et al. 2019, Huang
et al. 2019, Hernandez et al. 2021). Regarding drought stress, it has been proposed that positive
interactions should increase in frequency under stressful conditions, a response explained by the
stress gradient hypothesis (Bertness and Callaway 1994, Callaway et al. 2002, Hoek et al. 2016,
Velez et al. 2018, Hammarlund and Harcombe 2019, Piccardi et al. 2019).

On a more general note, the authors take the correlations as an indication of interactions,
however, correlations may derive from habitat-filtering or dispersal limitation processes, in
which case, inferring resilience and resistance may be less straightforward. Indeed, this may be
a possible reason for some of the results obtained by the authors which led to the rejection of
some of their hypotheses. In other words, if the authors were able to remove correlations that
were not due to habitat filtering, and particularly due to dispersal limitation, then the remaining
correlations may support their original hypotheses. Partitioning correlations due to dispersal
limitation, habitat filtering and interactions may not be possible in most cases, but for soil
samples (and perhaps rhizosphere), it may be possible if the authors have the spatial coordinates
for each sample (more details here: 10.1111/1755-0998.13079). At least the authors should
acknowledge the issue that correlations may be due to other processes than interactions.

10.1111/1755-0998.13079 Incorporating phylogenetic metrics to microbial co-occurrence
networks based on amplicon sequences to discern community assembly processes (Goberna et
al. 2019)

**Response: we agree with the reviewer that correlation does not equate with interaction but can**
**be ascribed to habitat-filtering or dispersal limitation. However, as pointed out by the reviewer,**
**it is not possible to remove the effect of habitat filtering in our case. Regarding dispersal**
**limitation, we used a taxon-taxon-space approach to find that only a small amount of network**
**links (0 – 2.94 %) is related to spatial distance (Fig. S13). These results are consistent with our**
**study in a homogenous, ploughed, one crop farmland.**

**Added text: Please see General Concern 5, above.**

In lines 176-179 the authors state that resistance is $1-R^2$ (when comparing control and droughted
communities) and resilience is $1-R$ (when comparing control and re-wetted communities).
However, in lines 180-186 (and in the figures and tables), the authors detail and discuss R^2 values,
rather than $1-R^2$. Perhaps this can be simplified? Since R^2 is related to the level of change
between treatments, perhaps the $1-R^2$ definition is not needed?

In Figure 1 is the significance indicated in every compartment for what comparison exactly? Could
the authors detail this in the legend? As the authors use the R^2 as a measure of resilience and
resistance, to claim for instance that “that the fungal mycobioime is more resistant than the
bacterial microbiome to both pre- and post- flowering drought”, it would be important to show
that these differences in R^2 between bacteria and fungi are significant.

**Response: We appreciate the reviewer’s suggestion and recognize the confusion caused by the**
**usage of $1-R^2$. Now we directly calculate resistance and resilience following the methods of Shade**
**et al 2012 and have removed the text about $1-R^2$ throughout our ms. As a result of this change,**
**we now use a T test to assess significance in the differences in resistance and resilience between**
**fungal and bacterial communities.**

**Added text: Please see General concern 7, above**

Other comments:

Line 248 "rewatered" should be re-wetted for consistency.

**Response: rewatered is replaced by rewetted throughout the manuscript.**

Line 258: Is network modularity determined by the number of modules detected?

**Response: No, the modularity is not solely determined by the number of modules, but also the**
**extent to which a module is separated from the other parts of the network. We added description**
**of modularity here.**

**Added: Modularity was defined as the measure of how much of the network is structured as**
**cohesive subgroups of nodes (modules) in which the density of interactions was higher within**
**subgroups than among subgroups.**

Line 341, what is -- for?

*Text in question: Limiting analyses of our resistance hypothesis H1 to networks of interactions*
*that are both significant and positive, we found an outcome similar to that seen for all interactions*
*– some combinations of compartment and stress showed support for H1 and others did not.*

**Revised: Signals of co-occurrence may be masked in all-correlation analyses that include**
**correlations that are both positive and negative, and both nonsignificant and significant.**
**However, when we limited the analyses of our resistance hypothesis H₁ to networks of**
**correlations that are both significant and positive, we found an outcome similar to that seen for**
**all correlations -- some combinations of compartment and stress showed support for H₁ and**
**others did not.**

Lines 467-469: were the p-values corrected for multiple testing? If not, why?

**Response: We agree with the reviewer's request for correcting for multiple testing. We now**
**correct p-values using the FDR method. Our key findings were not changed by the FDR correction**
**of p values. We have provided this information in our revised ms.**

**Added text: Please see General Concern 3, above**

**Reviewer #2 (Remarks to the Author):**

The ms describes a new analysis based on the recombination of two previously published
datasets that examines resistance and resilience of microbial communities (bacteria and fungi)
associated with leaves, roots, rhizosphere, and surrounding soil of an agricultural crop, sorghum,
subjected to drought stress. Using modern methodology (e.g., rDNA metabarcoding) the group
finds that drought disrupts the plant-associated microbial communities and that co-occurrence
networks among functional guilds of rhizosphere fungi and leaf bacteria were "dramatically
strengthened" in the pre-flowering drought treatment. The ms frames these finding within the
context of the classical stress gradient hypothesis, and also suggests that microbial 'hub' taxa

could be identified that might have utility as seed-taxa serving to support the microbial
communities overall under drought stress expected with climate change scenarios.

While I feel that the ms represents an important contribution to the field, especially considering
the cited deficits of previously published works that focus on agriculture, I feel that the ms is not
yet ready for publication. I recommend a significant revision that addresses the following
important issues:

>>Language awkwardness/precision/directness: I find the language of the ms to be very awkward
in several sections and in some areas the language also lacks precision and also could be edited
to be more straightforward. I present a few examples (primarily from the Introduction) here...

Introduction: starting @ line 57 (drought...drought/plant gene...plant genes) - "When drought
curtails photosynthesis in response to drought the most profound change in plant gene
transcription is the down regulation of plant genes involved in managing microbial association
and this change in expression correlates with a decline in the abundances of these root-
associated microbes." ... consider, "One of the most profound changes in plant transcription in
response to drought is the down regulation of genes involved in managing microbial association
that can result in a reduction in abundance of root-associated microbes."

**Response: we thank the reviewer for the suggested revision. However, it changed our original**
**meanings in two ways. First, for the rewording "one of the most profound changes". It was the**
**most profound change, that is, there was no more profound change in plant transcription in**
**response to drought than the down regulation of genes involved in managing microbial**
**association. Second, for 'that can result in ...', we found a correlation between those sorghum**
**genes and microbial abundance, but we cannot infer the direction.**

**We realize that our previous statement was confusing, and revised it as followed:**

**Revised: There is no more profound change in plant transcription in response to drought than**
**the down regulation of genes involved in managing microbial associations, and the down**
**regulation correlates with a reduction in abundance of root-associated microbes.**

Introduction: starting @ line 81 - "We surveyed previous research that included both fungi and
bacteria from the perspective of the community compositional response to drought and
subsequent rewetting (Table S1) finding that H1 has been both supported and falsified, and H2
has been either falsified or untested." ... consider, "We surveyed the literature for research that
addressed community composition shifts, for both fungi and bacteria, in response to drought and
subsequent rewetting (Table S1) finding that H1 has been both supported and refuted, while H2
has either been refuted or remains untested."

**We agree with the reviewer and have revised our text as suggested.**

Revised text: We surveyed the literature for research that addressed community composition
shifts, for both fungi and bacteria, in response to drought and subsequent rewetting (Table S1),
We find that H₁ has been both supported (Barnard et al. 2013, de Vries et al. 2018) and refuted
(de Vries and Shade 2013, McHugh et al. 2014, McHugh and Schwartz 2016), while H₂ has either
been refuted (de Vries and Shade 2013, de Vries et al. 2018) or remains untested (Barnard et al.
2013).

Introduction: starting @ line 96 - "Here, to advance our aim of including microbe-plant
interaction in efforts to combat crop loss due to drought, we test these hypotheses, H1 and H2,
through comparisons of microbial communities in four compartments (leaf, root, rhizosphere
and soil) in fields of sorghum during these three treatments, when drought imposed prior to
flowering, when this preflowering drought is relieved by watering, and when drought is imposed
after flowering." ... from my reading of the methods, this study was carried out during drought
conditions (i.e., it was not "imposed") in CA with crops being subjected to watering (rewetting)
or not, consider, "In this study we focused on both bacterial- and fungal-plant interaction,
examining hypotheses H1 and H2 for microbial communities associated with sorghum leaves,
roots, rhizosphere, and surrounding soils in agricultural fields under drought conditions that were
relieve post-flowering by watering or not." Further, while I agree that the results from this study
provide insights that might be helpful in efforts to "combat crop loss", the study did not directly
address "crop loss" and, therefore, statements such as this are likely a bit of an overreach.

Response: We thank the reviewer for the suggested revision, however, it failed to capture all
three treatments: 1) regular wetting throughout the season as a control, (2) pre-flowering
drought followed by regular wetting at flowering, (3) regular wetting before flowering that was
followed by post-flowering drought.

Note, this point is now moot because we have followed reviewer 3's suggestion to remove the
post-flowering drought treatment in this study.

We agree that drought in our study is not imposed. We also agree to remove the statement about
combat crop loss.

Revised: Here, we examine hypotheses H₁ and H₂ for microbial communities associated with
sorghum leaf, root, rhizosphere, and soil, in naturally droughted, agricultural fields experiencing
two irrigation treatments, (1) regular wetting throughout the season as a control, and (2) natural,
pre-flowering drought followed by regular wetting beginning at flowering.

Introduction: starting @ line 109 (Bacteria are typically considered a Domain, while Fungi are
typically considered a Kingdom) - "For example, regarding drought stress, it has been proposed
that positive interactions should increase in frequency under stressful conditions, a response
explained by the stress gradient hypothesis. It also has been proposed from studies of microbes
on Arabidopsis leaves, roots and soil, that correlations between microbes within kingdoms tend
to be positive, while correlations between kingdoms tend to be negative. Additionally, ecological
modeling has indicated that negative interactions should promote stability...." ...

consider, "For example, it has been proposed that positive microbial interactions should increase
in frequency under stress scenarios, such as drought, a response explained by the stress gradient
hypothesis (SGH). Further, stress studies of microbes on Arabidopsis leaves, roots, and the
surrounding soils suggest that within-taxonomic group microbial interactions tend to be positive,
while those between-taxonomic groups are negative. Ecological modeling also indicates...."
Further, microbial interactions, which biological/ecological in nature, should not be confused
with correlation, which is simply a statistical method. For example, positive correlations related
to shifting microbial abundances might be interpreted as mutualist interactions (or facilitation),
while negative correlations might be interpreted as antagonistic interactions (or competition).
The paper tends to confuse these concepts a bit (see comments immediately above and below),
and the authors should bear in mind that they are attempting to view/interpret microbial
interactions through the lens of statistical correlation (e.g., correlations metrics are appropriate
for the results, but the interpretation (i.e., in discussion) should focus on the interactions.

**Response: We thank the reviewer for the suggested revision of text as well as the interpretation**
**of the results regarding correlation and interaction. We accepted all these suggestion in**
**preparing the revised ms.**

**Added text: Please see General Concern 5, above.**

Introduction: starting @ line 112 - "*Using these studies to frame hypotheses at the all-correlation*
*level, for our resistance hypothesis, H1, under drought we expect an increase in the proportion of*
*positive correlation most strongly for B-B, followed by F-F, and lastly by B-F correlation; and for*
*our resilience hypothesis, H2, under re-watering, we expect a decrease in the proportion of*
*positive correlation most strongly for B-B, followed by F-F, and lastly by B-F correlation.*" This
sentence does not entirely make sense given the discussion as the proposed hypotheses are not:
516 A) clearly defined overall; B) completely consistent with the studies mentioned; or C) differently
defined for resistance vs. resilience - also, I'm not sure what phrases like "all-correlation level"
mean

...consider, "*These previous studies provide a framework for the hypotheses we propose here,*
*namely under the stress of drought, we expect enhanced facilitation within taxonomic groups (i.e.,*
*positive correlations for B-B and F-F) and enhanced competition between taxonomic groups (i.e.,*
*negative correlation for B-F).* Further, the hypotheses proposed by authors in the ms need to be
distinguished from those of other work (i.e., those associated with SGH) and more clearly defined
and consistent overall. For example, the hypotheses mentioned in the abstract focus on fungi
and state that fungi are "(i) more resistant but (ii) less resilient than bacteria" (we assume this
refers there respective status under the stress or drought), while the H1 and H2 mentioned here
focus on interactions.

**Response: Our hypotheses and the ways in which we evaluate them are a bit more complex than**
**presented by reviewer #2.**

We test our hypotheses that “fungi are (i) more resistant but (ii) less resilient than bacteria” at
three levels: a) using community composition, b) using all-correlations (we follow de Vries 2018),
and c) using just correlations limited to those that are significant and positive as determined from
a co-occurring network. In the part referred by reviewer #2, we focused on the test of these two
hypotheses at the all-correlation level.

Based on the stress-gradient hypothesis (stress increases frequency of positive microbial
interactions), the hypothesis that fungi will be more resistant than bacteria can be extended from
the community composition level to the all-correlation level. The expectation is that drought will
increase the proportion of positive correlation more strongly for B-B correlations than F-F
correlations. It is also possible to extend the Resilience hypothesis (Bacteria > Fungi) to the all-
correlation level, i.e., rewetting will decrease the proportion of positive correlations more
strongly for B-B correlations than F-F correlations.

The original framework for evaluating resistance (Fungi > Bacteria) and resilience (Bacteria >
Fungi) was limited to interactions within fungi or within bacteria and did not have expectation on
the interaction between bacteria and fungi (B-F). We added these inter-domain interactions
based on the results of research on *Arabidopsis* (within-taxonomic group microbial interactions
tend to be positive, while those between-taxonomic groups are negative) and ecological
modeling (negative interactions promote stability). In adding B-F interactions to resistance, we
hypothesized that drought would increase the proportion of positive correlation more strongly
for within-taxonomic group microbial interactions (B-B and F-F) than between-taxonomic groups
(B-F). In adding B-F interactions to resilience, we hypothesized that rewetting would decrease
the proportion of positive correlations more strongly for within-taxonomic group microbial
interactions (B-B and F-F) than between-taxonomic groups (B-F).

Putting all these items together, our resistance hypothesis is that “H₁, under drought we expect
an increase in the proportion of positive correlation most strongly for B-B, followed by F-F, and
lastly by B-F correlation”; and our resilience hypothesis is that “under rewetting, we expect a
decrease in the proportion of positive correlation most strongly for B-B, followed by F-F, and
lastly by B-F correlation”

We do not expect that “drought [would] enhance facilitation within taxonomic groups (i.e.,
positive correlations for B-B and F-F) and enhance competition between taxonomic groups (i.e.,
negative correlation for B-F).” Under the Stress Gradient Hypothesis, “drought [would] enhance
facilitation for both within taxonomic groups and between taxonomic groups”.

Here, we revise this paragraph to improve clarity regarding the resistance and resilience
hypotheses.

Revised: For example, it has been proposed that positive microbial interactions should increase
in frequency under stress scenarios, such as drought, a response explained by the stress gradient
hypothesis (SGH) (Bertness and Callaway 1994, Callaway et al. 2002, Hoek et al. 2016, Velez et al.
2018, Hammarlund and Harcombe 2019, Piccardi et al. 2019). Thus, when microbial correlations

among and between bacteria and fungi (all-correlation, B-B, F-F, B-F) are considered, if H₁ (fungi
are more resistant to drought stress than bacteria) is considered under the SGH, drought would
be expected to increase the proportion of positive correlations more strongly for B-B correlations
than F-F correlations, and if H₂ (fungi are less resilient to rewetting than bacteria) is similarly
considered, rewetting would be expected to decrease the proportion of positive correlations
more strongly for B-B correlations than F-F correlations. Although the original H₁ and H₂ were
based on bacteria or fungi, by themselves, and not interaction between bacteria and fungi,
interactions between bacteria and fungi were included in two more recent studies. First, stress
studies of microbes on *Arabidopsis* leaves, roots, and the surrounding soils indicated that within-
taxonomic group microbial interactions tended to be positive, while those between-taxonomic
groups were negative (Agler et al. 2016, Duran et al. 2018). Second, ecological modeling indicated
that negative interactions should promote stability of communities (Coyte et al. 2015). Therefore,
using these studies to frame hypotheses focusing on all-correlations, for our resistance
hypothesis, H₁, under drought we expect an increase in the proportion of positive correlation
most strongly for B-B, followed by F-F, and lastly by B-F correlation; and for our resilience
hypothesis, H₂, under rewetting, we expect a decrease in the proportion of positive correlation
most strongly for B-B, followed by F-F, and lastly by B-F correlation.

Introduction: paragraph @ line 118-136 - I find this paragraph to be confusing and repetitive with
respect to the hypotheses (and see above) overall, the discussion of "nonintuitive outcomes" is
a bit obtuse and appears to be splitting hairs (to justify results/methods?). Also, "Simplifying
matters by focusing on just the significant, positive correlations" - if a correlation is not significant
then it should not be considered as a result at all; further, the paragraph above and H1/H2 stress
the importance of validating negative correlations. This paragraph appears to be justification for
the methods used in the co-occurrence network analysis part of the study, but the case could be
more clearly and directly made (i.e., this is a common method for such analyses).

**Response: We agree with the reviewer and simplified this paragraph. We follow the approach of**
**de Vries by including both significant and non-significant correlations in all-correlation analysis,**
**and only significant, positive correlations in the co-occurrence network.**

*Revised: Integrating positive with negative correlations can lead to nonintuitive outcomes, for*
*example, if both positive and negative interactions decrease, the sum can be positive if the*
*decrease is strongest for the negative correlations. Simplifying matters by focusing on just the*
*significant, positive correlations has revealed new information on co-oscillation of microbial taxa*
*and the stability of communities (de Vries et al. 2018). Co-occurrence network analysis focuses*
*on co-oscillation of microbial taxa in response to perturbation (de Vries et al. 2018). That is, it*
*focuses on just the significant, positive interactions.*

Introduction: starting @ line 137 - The authors should note different terminology typically use in
distinguishing between network element vs. network properties. For example, 'modules' are
network elements (functional units of connectedness within the network) whereas 'modularity'
is a network property (the characteristic of being divided into multiple modules); likewise, 'hubs'

are network elements (nodes with a number of links/edges that greatly exceeds the average) and
'hub emergence' (networks that reflect the characteristic of contain multiple highly linked hubs).

**Response: we agree with the reviewer and have replaced the word 'properties' with 'elements'**
**Revised:** Identification of key network **elements**, such as, modules or hubs, may facilitate
practical application of microbial networks to modern agriculture

Introduction: starting @ line 148 - "Our experimental system is an agricultural field....Compared
to previous studies, our system is simpler because it has just one plant genotype, which is grown
in synchrony...Our identification of bacteria and fungi by DNA sequence is more precise...." Etc....
rather than directly comparing the work carried out here to previous studies, it might be more
preferable to simply state the strengths of this study (the relative improvement over earlier work
should already be clear from justifications provided in previous chapters within the Introduction),
consider "Here we use modern high-throughput sequencing techniques to examine interactions
of microbial communities, bacterial and fungal, associated with leaves, roots, rhizosphere, and
surrounding soils of two sorghum cultivars planted as a monocultures in agricultural fields during
a period of drought. This experimental system allowed us to investigate resistance and resilience
of these microbial communities under the stress of drought and subsequent recovery after
watering...etc."

**Response: We agree with the reviewer and have revised this paragraph following the reviewer's**
**suggestion.**

**Revised:** Here, we address the hypotheses about resistance (H_1) and resilience (H_2) using three
approaches, (i) whole community composition, (ii) all pairwise correlations among individual taxa,
and (iii) the co-occurrence network of significant positive interactions. In a semiarid agricultural
field where control plots were watered regularly and test plots were **naturally** droughted before
flowering **followed by regular wetting beginning at flowering**(Xu et al. 2018, Varoquaux et al.
2019, Gao et al. 2020), we used modern high-throughput sequencing techniques to examine
communities of bacteria and fungi associated with leaf, root, rhizosphere, and surrounding soil
of two sorghum cultivars planted as a monocultures during a growing season. **One might wonder**
**if the microbes in these fields were already adapted to drought, however a six-decade history of**
**irrigated agriculture at the site indicates that the microbes in our system are not drought adapted**
**(Gao et al. 2020).** Thus, this experimental system allowed us to investigate resistance and
resilience of these microbial communities under the stress of drought and subsequent recovery
after watering. Community assembly of both fungal mycobiome and bacterial microbiome were
published earlier in separate papers (Xu et al. 2018, Gao et al. 2020). Here, we newly analyzed
these two datasets together to test H_1 and H_2 using **the** three approaches **noted** above.

>>Questions related to approach, interpretation, and statistics used:

Ecological concepts: The authors state, "We use definitions of ecological resistance as the change
in compositional dissimilarity in response to stress and of ecological resilience as the recovery in

compositional dissimilarity when stress is relieved. Ecological resistance and resilience are
determined by comparing compositional dissimilarity among communities within treatments
(combined control and stress) with dissimilarity between control and stress communities.
Specifically, resistance is $1-R^2$ using control and droughted communities and resilience is $1-R^2$
using control and rewetted communities, in which R^2 was determined by permutational analysis
of variance (permanova 40)." The authors should directly cite works influencing the definitions
here, for example the referenced paper Shade et al. 2012 provides excellent discussion over the
concepts of resistance and resilience as well as related terminology. These authors state,
"Disturbance and community stability are necessarily related, as stability is defined as a
community's response to disturbance (Rykiel, 1985). Here, we adopt definitions most similar to
Pimm (1984), in which stability is comprised of resistance and resilience (Table 1), two
quantifiable metrics that are useful for comparing community disturbance responses and have
precedent in the microbial ecology literature (e.g., Allison and Martiny, 2008). ... Here, resistance
is defined as the degree to which a community is insensitive to a disturbance, and resilience is
the rate at which a community returns to a pre-disturbance condition (Pimm, 1984)." These
authors further define the related 'Stable state' as, "A condition where a community returns to
its original composition or function following disturbance." As the ms authors base their analyses
on Bray-Curtis dissimilarity, it should be noted that here a value of 0 means two sites that have
the same community composition (they share the same species at the same levels of abundance),
whereas a value of 1 denotes two communities that are completely dissimilar (i.e., they do not
have species in common). Given this, could resilience, for example, be better defined as "the
recovery in compositional similarity (i.e., Bray-Curtis dissimilarity values converging on zero)."
Such a definition would have bearing, for example, on the interpretation of Figure 1. Further, this
figure also stresses the reliance on the R-squared value (inversely proportional to the effect
strength) in interpreting resistance or resilience, yet the generally low R^2 here suggests very little
variation in distances is explained by the groupings - are we to believe that this means (inversely)
very strong resistance or resilience effects? Further, p-values in Permanova type are strongly
influenced by sample size, was this accounted for in the analysis (similarly see comments
regarding FDR below). Some of these issues need to be cleared up.

**Response: We appreciate the reviewer's suggestion and realized the confusion caused by the**
**usage of $1-R^2$. Now we directly calculated resistance and resilience following the methods of**
**Shade et al 2012 and removed the part about $1-R^2$ throughout our ms.**

**Added text: Please see General Concern 7, above.**

Network analysis: When running numerous parallel correlations, as are possible with
metabarcoding sequence data, the chance of recovering spurious significant positive correlations
are greatly enhanced. There are statistical methods, such as FDR (false discovery rate), that can be
used to reduce the influence of false positives. This may be especially true for non-parametric
approaches (i.e., Spearman's ranked correlation). Corrective measures (i.e., FDR), may be
warranted here to reduce type I errors.

Response: We agree with the reviewer. Now, the p-values are corrected for multiple testing using
the FDR method. Our key findings were not changed by the FDR correction of p values. We added
this information in the revised manuscript.

Revised text: Please see General concern 3, above.

Guild approach: The author also use a fungal guild concept in their network analyses, while these
concepts appear to be derived from the paper below, yet the authors do not directly cite this
paper/source/software and should (especially in the methods):

Response: We thank the reviewer for detecting our omission. This reference is now cited in our
revised manuscript.

Nguyen NH, Song Z, Bates ST, Branco, S, Tedersoo L, Menke J, Schilling JS, Kennedy PG. 2016.
FUNGuild: an open annotation tool for parsing fungal community datasets by ecological guild.
Fungal Ecology 20:241-248.

Further, care should also be taken when interpreting the network analysis results. For example,
the authors claim that "co-occurrence networks among functional guilds of rhizosphere fungi ...
were dramatically strengthened by pre-flowering drought", yet Figure 2B show that the
"strengthened" network contains numerous saprotrophs and plant pathogens, suggesting that
"pre-flowering drought" contributed to decay (perhaps of dead plant matter) and disease.

Response: In this study, we found that a number of fungal pathogens are present in the network,
that these fungi correlate with saprotrophic, endophytic and mycorrhizal fungi, and that
correlations among fungal OTUs increased. However, in our previous analysis of fungal
community composition (Nat Comm paper, Fig. 5A, C; Fig. S2A), we showed that pre-flowering
drought drastically reduced the relative abundance of fungal pathogens. Thus, although
correlation between fungal OTUs increased, it is not likely that plant disease or decay increased
in pre-flowering drought.

Added: The strengthened fungal network in rhizosphere seen in this study was coupled with the
co-occurrence of a number of fungal pathogens with saprotrophic, endophytic and mycorrhizal
fungi. However, it is not likely that there was an increase in plant decay or disease, because we
previously found that the relative abundance of rhizosphere fungal pathogens was drastically
decreased by pre-flowering drought (Gao et al. 2020).

Also see comments above regarding potential overreaching statements.

Examples of other issues:

Introduction: starting @ line 153 - "seedling emergence to fruit maturation" ... as sorghum is a
member of the Poaceae (i.e., a grass) the seed (e.g., millet) of sorghum is typically referred to as
a cereal grain rather than a "fruit".

Response: We agree with the reviewer that the seeds of grasses, like sorghum, are typically
referred to as grains. However, we feel that we are botanically correct in that sorghum, like all
angiosperms, makes a fruit with seed surrounded by a fruit or pericarp. In the case of grasses,
the pericarp is fused to the seed coat and is termed a caryopsis.

However, we deleted this sentence in light of this reviewer's other comment.

Results: starting @ line 165 - "As noted above, the simple fact that fungi grow more slowly than
bacteria..." I don't feel that this is a simple matter, bacteria "grow" as single-celled
microorganisms through binary fission where, yes, doubling times can range in 10s of mins.
Growth for fungi is something completely different; a (sometimes massive) multicellular
(generally) mass of hyphae (a mycelium) that grows by extension at the hyphal tip (unless we are
talking about yeasts), where some taxa (e.g., Neurospora) can have relatively high growth rates
(e.g., several mm per hour) at the hyphal tip. Therefore, a reductionist approach to growth rates
is likely not warranted here.

Response: Although we agree with the reviewer that some fungi can grow quickly and some
bacteria can grow slowly, it is generally accepted that most fungi grow more slowly than bacteria.

Because the concept that fungi respond more slowly than bacteria to stress is fundamental to
the resistance/resilience hypothesis as developed by de Vries and Shade, we feel that it should
remain in the manuscript.

Revised text: As noted above, the simple fact that fungi grow more slowly than bacteria is the
basis of the hypotheses that (H₁) fungal communities should be more resistant than bacterial
communities to drought stress, and (H₂) that fungal communities should be less resilient than
bacterial communities when the stress is relieved by rewetting (de Vries and Shade 2013). In
addition to growth rate, these two hypotheses may be related to differences in growth form
between fungi and bacteria. For example, multicellular hyphal growth versus unicellular division
or the greater thickness of fungal cell walls as compared to those of bacteria (Schimel et al. 2007,
Guhr et al. 2015).

Results: paragraph @ lines 165-179 - There are no results given here, this paragraph has
elements that may be more appropriate for the Methods section.

Original text: As noted above, the simple fact that fungi grow more slowly than bacteria is the
basis of the hypotheses that (H₁) fungal communities should be more resistant than bacterial
communities to drought stress, and (H₂) that fungal communities should be less resilient than
bacterial communities when the stress is relieved by rewetting 18. We tested these hypotheses
at the community composition level by blending the fungal and bacterial datasets generated
from the same leaf, root, rhizosphere and soil samples collected from field-grown sorghum that
had been either irrigated as a control, or subjected to pre-flowering drought or post-flowering
drought 10,11. We use definitions of ecological resistance as the change in compositional
dissimilarity in response to stress and of ecological resilience as the recovery in compositional

dissimilarity when stress is relieved. Ecological resistance and resilience are determined by
comparing compositional dissimilarity among communities within treatments (combined control
and stress) with dissimilarity between control and stress communities. Specifically, resistance is
1-R2 using control and droughted communities and resilience is 1-R2 using control and rewetted
communities, in which R2 was determined by permutational analysis of variance (permanova 40).

**Response: we added text and figure, please see General Concern 7, above.**

**Reviewer #3 (Remarks to the Author):**

In this manuscript, the authors report the effect of pre-flowering drought, post-flowering drought,
and recovery after pre-flowering drought on fungal and bacterial communities and networks
in/on roots, rhizosphere soil, bulk soil, and leaves of field-grown sorghum. They hypothesise,
based on previous work, that fungal communities and network are more resistant but less
resilient than those of bacteria. They test these hypotheses using previously published data for
new analyses. They find that their hypothesis that fungal communities are more resistant and
less resilient than bacterial communities is supported. Using all correlations between bacteria
and fungi in the four compartments, they find that the frequency of positive correlations
increased in pre-flowering drought, but using only significant positive correlations (ie co-
occurrence networks), they find that pre-flowering drought disrupts networks in roots,
rhizosphere and soil but increases their connectivity on leaves. Re-watering resulted in networks
resembling control networks again, except for the network in soil (but note that I inferred those
results myself from Fig. 3 as I found the description of the results hard to follow). They conclude
that understanding microbial network response to stress might inform manipulating microbial
communities for increased plant tolerance to stress in agricultural settings.

I enjoyed reading this mostly clearly written manuscript that addresses interesting hypotheses.
However, I found the amount of results presented quite overwhelming and not always easy to
follow/ interpret. The hypotheses stated are quite abstract and informed entirely by previous
work on soil fungal and bacterial communities and network responses to drought, and in that
sense the paper reads as largely confirmatory and leans heavily on the results from a few recent
papers. I also feel that there is really a severe lack of context on why we want to understand how
the communities/ networks in these different plant compartments respond to drought. To me, it
would be much more interesting to focus in on the differences between these compartments.
What drives the assembly of fungal and bacterial communities on leaves, and how is this different
from those in roots and in soil? What would be the implications for their functioning and for plant
health of the changes in these communities in response to drought? I am missing all of this in the
manuscript, other than quite vague and general statements.

**Response: We agree with the reviewer that understanding the drivers of community assembly in**
**different compartments is an interesting topic and, in fact, we have investigated this topic in our**
**previous published studies (Gao et al 2020 Nat Com; Xu et al 2018 PNAS). Those studies focused**
**on fungi and bacteria independently. Here, we compare fungi and bacteria and examine their co-**
**occurrences. In particular, we investigate the resistance and resilience of bacterial and fungal**

communities. We feel that this question is of broad interest to all ecologists and are encouraged
that all four reviewers' comment on the importance of this topic.

We hesitate to add more information about the context of different compartments, with this
modest exception.

Revised: In the interior and surface of different compartments such as leaf, root and rhizosphere,
crop plants form essential beneficial partnerships with microbes, both fungi and bacteria, that
impact plant drought responses.

I would suggest to focus on this, and I would also suggest ditching the post-flowering drought
treatment, as there is no recovery phase after this drought, which makes it difficult to compare
these data to the pre-flowering drought.

Response: We appreciate the reviewer's suggestion to focus on pre-flowering drought. We agree
with the reviewer and have removed the part on post-flowering drought in the new ms.

Added text: The experimental design of pre-flowering drought followed by regular wetting
beginning at flowering represent an ideal system for testing the hypotheses that fungi are (i)
more resistant to drought stress but (ii) less resilient when the stress is relieved by rewetting than
bacteria. However, the experimental design of regularly watering followed by post-flowering
drought is not relevant to these two hypotheses. Therefore, for simplicity, this study only
included control and pre-flowering drought (followed by rewetting) treatments and did not
analyze the post-flowering drought treatment.

Moreover, while the manuscript focusses on networks, never is the reliability of these
correlations and whether they actually represent interactions between microbes discussed.
Positive correlations between microbes can simply indicate niche sharing or responding to the
same drivers. Moreover, it is not clear which OTUs were used for correlations (all? Or the ones
that occurred over a certain number of experimental units? Or the most abundant ones?), and
on how many observations these correlations are based. From the methods it seems that there
were 6 replicates of each treatment – does this mean that correlations were based on only 6 data
points? Then I would seriously question the robustness of the resulting networks.

Response: We agree with the reviewer that correlation does not necessarily mean interaction.
We now discuss inferring microbial interaction from microbial correlation.

Revised text, please see General Concern 5 above.

We thank the reviewer for letting us know that our OTU selection was not clear. Not all OTUs are
used for correlation analysis, we only used taxa with > 30 reads and occurred in at least 8
communities in each analysis. In this regard, we are following the approach of Shi et al (Shi et al.
2016) and de Vries et al (de Vries et al. 2018).

We thank the reviewer for alerting us to the fact that the number of data points was obscure.
Correlations for network analyses were not limited to 6 data points, rather we used 36 or 48 data
points. Our analyses combine several different time points for the same treatment. For drought,
we had 36 data points (6 plots * 6 time points = 36 data points) and for re-wetting we had 48
data points (6 plots* 8 time points = 48 data points).

Added text: Please see General Concern 4, above

In addition, while on close inspection the analyses seem robust and the results are mostly
correctly interpreted, I found the figures quite hard to understand as the axes and legends are
rather ambiguous. The clarity can be improved, and perhaps also the presentation, because as I
said above the amount of data is overwhelming.

Response: Again, we thank the reviewer for alerting us to the difficulty interpreting figures. We
believe that we have improved the presentation and legends of all the figures.

More detailed comments:

L 164: yes, but also because of their hyphal growth form and thick cell walls, see Schimel et al.
2007 Ecology and Guhr et al. 2015 PNAS.

Response. We agree with the reviewer and have revised our text.

Revised text: In addition to growth rate, these two hypotheses may be related to differences in
growth form between fungi and bacteria. For example, multicellular hyphal growth v. unicellular
division or the greater thickness of fungal cell walls as compared to those of bacteria (Schimel et
al. 2007, Guhr et al. 2015).

L 175-184 and Figure 1: I found this section very hard to follow. Here, it says that resistance and
resilience are calculated as 1-R2, but in the figure Bray-Curtis dissimilarities are reported (are
similarities? This is not clear), and in the figure legend it says resistance and resilience. I am lost.
It's also not immediately clear what is meant by inter-group and intra-group.

Response: We realized the confusion caused by the usage of 1-R2. Now we directly calculated
resistance and resilience following the methods of Shade et al 2012, and removed the part about
1-R2 throughout our ms.

Added text: Please see General Concern 7, above.

L 205: can you be more specific? Which compartments?

Text in original ms: Neither did we find consistent support for the differences ascribed to bacteria
and fungi in H₂ as the strongest decreases in the proportion of positive correlations during
rewetting could occur in any of the three comparisons (F-F in rhizosphere and soil, B-B in root,
and B-F in leaf) (Fig. 2B).

**Response: We agree with the reviewer that we could be more specific.**

**Revised text: Neither did we find consistent support for the differences ascribed to bacteria and**
**fungi in H₂ as the strongest decreases in the proportion of positive correlations during rewetting**
**occurred at F-F in rhizosphere and soil, and B-B in leaf and root (Fig. 2B).**

L 238-244: I found this section very hard to read, as pretty much every sentence mentions that
vertices are dropped and rise, but in response to what and compared to what? I assume to
drought, but this is never explicitly mentioned.

Text in original ms: In general, for pre-flowering drought, we found no consistent support for the
difference between bacteria and fungi inherent in H₁. Rhizosphere was the one compartment
where B-B vertices dropped and F-F vertices rose, as expected, but was offset by root and soil,
where vertices dropped in all networks, B-B, F-F and B-F (Fig. 3-4; Fig. S2-4). In leaf, the result was
the opposite of expectation, as B-B rose while F-F was unchanged.

**Response: We have attempted to simplify a complex result, below.**

**Revised: In general, we found no consistent support for the difference between bacteria and**
**fungi inherent in H₁. Rhizosphere was the one compartment where B-B vertices dropped and F-F**
**vertices rose in response to drought, as expected, but this result was offset in root and soil, where**
**vertices dropped in all networks, B-B, F-F and B-F (Fig. 3-4; Fig. S3-4).**

L 252: The biotic interactions become even more complex than the control after rewatering. But
is this resilience? Resilience means that the disturbed treatment is approaching or resembling
the control.

Text in original ms: However, we found no support for the H₂ in leaf and root where the F-F did
not lose complexity, although both the B-B and B-F networks gained complexity (Fig. 3-4, Fig. S2,
S3).

**Response: We appreciate the reviewer for pointing out this complexity. Our results suggest that**
**resilience does not necessarily stop when approaching the control values, but that resilience can**
**exceed the control. This situation has rarely been observed, but we find it in our results.**

**Added text: Our results suggest that resilience does not necessarily stop when approaching the**
**control values, but that resilience of biotic interaction can exceed the control. Our data highlight**
**a phenomenon that has rarely been reported (Shade et al. 2012).**

L 315-318: I don't understand this sentence

Text in original ms: The main difference between our study and these others is our use of one
species of plant whose growth is synchronous whereas none of the other studies focused on just
one plant species [although de Vries, et al. 19 used just 4 plant species]. Other salient differences
include our using DNA sequence of variable regions to identify bacteria and fungi and our field

season being free of precipitation, making it straightforward to impose drought and then relieve
it through irrigation.

**Response: We have attempted to make the sentence more understandable.**

**Revised text: The main difference between our study and these others is the simplicity of our**
**system, the use of DNA metabarcoding to identify microbes and the dependability of natural**
**drought in an arid environment. We used just one species of plant whose growth is synchronous**
**whereas all other studies focused on at least four species (de Vries et al. 2018) and typically many**
**plant species. We used DNA sequence of variable regions to identify bacteria and fungi. Our field**
**season was free of precipitation for the entire growing season, making it straightforward to**
**experience drought and then relieve it through irrigation.**

L 325: not just in leaf in post-flowering drought, also in soil and root

L 324: De Vries et al. 2018 Nat Comms also analysed combined bacterial-fungal networks – this
is detailed in their supplementary material

Text in original ms: Extending the analysis to previously unexamined B-F interactions, we found
increases in all compartments except soil in the pre-flowering drought and leaf in post-flowering
drought (Fig. 2).

**Response: We thank the reviewer for pointing out these facts. we rephrased this sentence**
**accordingly.**

**Revised: Extending the analysis to previously poorly examined B-F interactions, we found**
**increases in these interactions in root and rhizosphere but no change in soil (Fig. 2).**

L 327-330: this sentence makes no sense to me. Hypotheses developed from one type of analysis?
I would think that it is not about the analysis but about the concept. The analysis is just a means
to test a hypothesis.

Text in original ms: A simple explanation for our observations is that hypotheses developed from
one type of analysis are specific to that type, and that empirical hypotheses are more difficult to
reject than those based on models.

**Response: We removed this sentence.**

L330-331: again, I have no idea what is meant here. Whole communities hide variation based on
compartments?

Text in original ms: What is also clear is that analyses of whole communities hide variation based
on compartment as well as the identities of partners in particular interactions.

**Response: We removed this sentence.**

L332-334: I think it is rather stark to make inferences about applications in agriculture from these
theoretical hypotheses

Text in original ms: These two aspects will be important to efforts to manipulate microbes to
improve agricultural outcomes because effective application of microbes to affect agricultural
outcomes must involve specific microbes and compartments.

Response: **We removed this sentence.**

Methods: I understand that these are previously published data but there's really more detail
needed here. How large were the plots? What was the experimental layout? How were samples
collected? What other analyses were done? Were there six replicates per treatment, and does
this mean that correlations for network analyses were done only using 6 datapoints....?

Response: **we now provided more info about the experiment design and sampling, and data
analysis.**

Added texts: **Please see General concern 4, above.**

**Reviewer #4 (Remarks to the Author):**

Cheng Gao and colleagues in their manuscript 'Resistance and Resilience in Microbes: Co-
occurrence Networks Delve Deeper Than Community Composition' address two fundamental
questions in the field of microbial community compositions: Resistance and resilience. To do so,
they combine two very comprehensive previously published datasets analysing microbial
communities on crop plants under extreme drought conditions and irrigation. The datasets are
based on 16S and ITS amplicon sequencing and the analyses in the paper is primarily based on
pairwise correlations of these datasets.

Particularly the question if fungi are more resistant H1 but less resilient H2 than bacteria is
certainly a key question in the field and addressed in depth in this manuscript. Besides direct
analyses of correlation data, the authors use networks to get deeper insights into community
structures. They identify a disruption of communities by drought and see an increase of positive
correlations among bacteria, fungi and across kingdoms correlating bacteria and fungi. In
combination with network analyses, this gives support for the stress gradient hypothesis. Based
on their analyses, they can further underpin the importance of mycorrhiza fungi in stabilizing
communities under drought.

In summary, the paper touches a very timely and relevant field and the authors show convincingly
that their dataset can be used to infer their central hypothesis H1 and H2. Although I think this
manuscript has great potential it would certainly benefit from more details and by addressing
some of the following points:

1. As the authors state, key to the paper are pairwise correlations. The authors focus, however,
only on Spearman's Rho or Spearman's rank-order correlation. This assumes a monotonic
relationship. From the paper it is not clear if the authors have analyzed other correlations to show
that this fits the best or have plotted the data to see if this really fits for all samples. Why not
using Spearman's correlation, particularly for the networks this might be a better choice or a
combination?

Response: we used Spearman's correlation. We guess the reviewer asks about our not using
Pearson's correlation.

Response: We used Spearman correlation to make our work comparable with the study of de
Vries et al 2018 Nat Commun, who also used Spearman correlation.

We added Pearson correlation and found a similar pattern of networks. We now provide this
result as a supplementary Figure. We also added the CoDa approach, as described in response to
the next comment by Reviewer #4.

Added text and figures. Please see response to General Concern 2, above.

2. Further to the correlation analyses: How valid is it to correlate 16S and ITS data together to
make conclusions about robustness and resilience? Both will result in completely different
resolution. ITS is used to resolve on a species level, 16S will rarely branch that deep. Wouldn't it
be better to compare 16S and 18S? Is it possible that bacteria are more resilient because of less
resolution, meaning other bacteria move in following rewetting but they are seen as having the
same 16S sequence while fungi move back in that show the same taxonomic distance but can be
resolved?

Response: Thank you for pointing out this concern. We are aware about the reviewer 4's concern
that 16S and ITS identify bacteria and fungi at different levels of taxonomic resolution (Bruns &
Taylor 2016 Science). However, we feel that lessening the resolution for fungi will not help the
analyses. Raising the resolution for bacteria would help the analyses, but we, and all other
researchers, are limited at the present to 16S for bacterial identification.

It is not clear to us that microbial communities might appear more resilient when more coarsely
identified. For example, if all fungi were sorted into two phyla, Ascomycota and Basidiomycota,
it would be very difficult to detect either resistance or resilience. As taxonomic identification
became more finely determined, resistance and resilience could be discerned. However, it is not
clear that the response to stress or its relief would be favored as taxonomic determination
became increasingly refined.

Still, to relief the reviewer's concern about the different resolution of 16S and ITS, we compared
bacterial 16S OTUs against both fungal ITS OTUs as well as fungal families. We reported only
results that are robust across these two conditions.

Added text and figures: Please seen response to General concern 1, above.

3. Very much depends on the calculation of the networks. From the methods I can see igraph has
been used and the implemented calculation of networks. To better understand the quality and
robustness of the networks it certainly needs more information on the calculation. For example,
how was sparsity addressed and how density of the networks. Based on the figures, density is a
particular issue, as very dense networks are compared to extremely sparse networks. I would

suggest to use at least one other method to calculate the networks correcting for abundance and
sparsity or not correcting and comparing those to each other. In my opinion this is relevant to
identify if modularity is robust, as this has been debated a lot.

**Response:** In addition to the Spearman and Pearson method, we made additional network
analyses using the CoDa method of Gloor et al 2017 to account for the sparsity of the data. We
only report the results that are robust across these three methods. These three methods showed
similar patterns in terms of the difference between control and drought, and between control
and rewetting.

We keep the results of Spearman in the main figures, as Spearman method is widely used in
ecological research such as de Vries et al 2018 Nat Commun. Also we keep the result of Pearson
and CoDa method in the supplementary.

We now provide more information about the calculation of the networks.

**Added text and figures:** Please see response to General Concern 2, above.

4. As far as I understand from the data sets, the samples are not independent from each other
but have a time factor: PRE-Drought, PRE-Rewatering, POST-Drought. To analyze stability it would
be useful to track vertices over time and compare PRE and POST networks directly. Particularly
positional stability of each vertex would be a good additional measure when comparing different
network calculations.

**Response:** Although it is desirable to track vertices over time, we are unable to do so because we
have six replications in each time point and would need at least ten replicates for this analysis.
We do note that six replicates at each time point is twice the norm in studies of microbial
communities.

5. A minor thing but relevant to understand what has been done: What are the Guilds and how
have they been calculated? I guess this is based on Nguyen et al 2016 but I could not find any
information.

**Response:** We now cite Nguyen et al 2016 in the our revised manuscript.

6. Question concerning the experimental layout: The experiments have been set up in an area
with extremely low precipitation. So any microbe in the soil would be adapted to cope with
drought. In this case I would assume that regular irrigation is a perturbation to the community
and not drought. Have samples been taken before the planting that could be compared? Is the
drought state perhaps a communal 'recovery'?

**Response:** The reviewer raises an interesting point. Although the precipitation is low in our
research area in the Central valley, our site has been in agricultural cultivation with irrigation for
more than 60 years. We have thought about this question quite a bit and our thinking is that our
microbes are likely adapted to irrigation and that the perturbation is drought is perturbation.

Added text: One might wonder if the microbes in these fields were already adapted to drought,
however a six-decade history of irrigated agriculture at the site indicates that the microbes in our
system are not drought adapted.

Literature cited in this response to reviewers.

Agler, M. T., J. Ruhe, S. Kroll, C. Morhenn, S. T. Kim, D. Weigel, and E. M. Kemen. 2016.

Microbial Hub Taxa Link Host and Abiotic Factors to Plant Microbiome Variation. *PLoS*
*Biol* **14**:e1002352.

Barnard, R. L., C. A. Osborne, and M. K. Firestone. 2013. Responses of soil bacterial and fungal
communities to extreme desiccation and rewetting. *Isme Journal* **7**:2229-2241.

Bertness, M. D., and R. Callaway. 1994. Positive interactions in communities. *Trends Ecol Evol*
**9**:191-193.

Callaway, R. M., R. W. Brooker, P. Choler, Z. Kikvidze, C. J. Lortie, R. Michalet, L. Paolini, F. I.

Pugnaire, B. Newingham, E. T. Aschehoug, C. Armas, D. Kikodze, and B. J. Cook. 2002.

Positive interactions among alpine plants increase with stress. *Nature* **417**:844-848.

Coyte, K. Z., J. Schluter, and K. R. Foster. 2015. The ecology of the microbiome: Networks,
competition, and stability. *Science* **350**:663-666.

de Vries, F. T., R. I. Griffiths, M. Bailey, H. Craig, M. Girlanda, H. S. Gweon, S. Hallin, A.

Kaisermann, A. M. Keith, M. Kretzschmar, P. Lemanceau, E. Lumini, K. E. Mason, A.

Oliver, N. Ostle, J. I. Prosser, C. Thion, B. Thomson, and R. D. Bardgett. 2018. Soil

bacterial networks are less stable under drought than fungal networks. *Nat Commun*
**9**:3033.

de Vries, F. T., and A. Shade. 2013. Controls on soil microbial community stability under climate
change. *Front Microbiol* **4**:265.

Dini-Andreote, F., M. de Cássia Pereira e Silva, X. Triadó-Margarit, E. O. Casamayor, J. D. van

Elsas, and J. F. Salles. 2014. Dynamics of bacterial community succession in a salt marsh

chronosequence: evidences for temporal niche partitioning. *The Isme Journal* **8**:1989-
2001.

Duran, P., T. Thiergart, R. Garrido-Oter, M. Agler, E. Kemen, P. Schulze-Lefert, and S. Hacquard.

2018. Microbial Interkingdom Interactions in Roots Promote Arabidopsis Survival. *Cell*
**175**:973-983 e914.

Farrer, E. C., D. L. Porazinska, M. J. Spasojevic, A. J. King, C. P. Bueno de Mesquita, S. A. Sartwell,

1177 J. G. Smith, C. T. White, S. K. Schmidt, and K. N. Suding. 2019. Soil Microbial Networks

Shift Across a High-Elevation Successional Gradient. *Frontiers in Microbiology* **10**.

Faust, K., G. Lima-Mendez, J.-S. Lerat, J. F. Sathirapongsasuti, R. Knight, C. Huttenhower, T.

Lenaerts, and J. Raes. 2015. Cross-biome comparison of microbial association networks.

*Frontiers in Microbiology* **6**.

Gao, C., L. Montoya, L. Xu, M. Madera, J. Hollingsworth, E. Purdom, V. Singan, J. Vogel, R. B.

Hutmacher, J. A. Dahlberg, D. Coleman-Derr, P. G. Lemaux, and J. W. Taylor. 2020.

Fungal community assembly in drought-stressed sorghum shows stochasticity, selection,
and universal ecological dynamics. *Nat Commun* **11**:34.

Goberna, M., A. Montesinos-Navarro, A. Valiente-Banuet, Y. Colin, A. Gómez-Fernández, S.

Donat, J. A. Navarro-Cano, and M. Verdú. 2019. Incorporating phylogenetic metrics to

microbial co-occurrence networks based on amplicon sequences to discern community
assembly processes. *Molecular Ecology Resources* **19**:1552-1564.

Guhr, A., W. Borken, M. Spohn, and E. Matzner. 2015. Redistribution of soil water by a
saprotrophic fungus enhances carbon mineralization. *Proceedings of the National*
*Academy of Sciences* **112**:14647-14651.

Hammarlund, S. P., and W. R. Harcombe. 2019. Refining the stress gradient hypothesis in a
microbial community. *Proc Natl Acad Sci U S A* **116**:15760-15762.

Hernandez, D. J., A. S. David, E. S. Menges, C. A. Searcy, and M. E. Afkhami. 2021. Environmental
stress destabilizes microbial networks. *The Isme Journal* **15**:1722-1734.

Hoek, T. A., K. Axelrod, T. Biancalani, E. A. Yurtsev, J. Liu, and J. Gore. 2016. Resource
Availability Modulates the Cooperative and Competitive Nature of a Microbial Cross-
Feeding Mutualism. *PLoS Biol* **14**:e1002540.

Huang, R., S. P. McGrath, P. R. Hirsch, I. M. Clark, J. Storkey, L. Wu, J. Zhou, and Y. Liang. 2019.
Plant-microbe networks in soil are weakened by century-long use of inorganic
fertilizers. *Microbial Biotechnology* **12**:1464-1475.

Jiang, Y., Y. Lei, Y. Yang, H. Korpelainen, Ü. Niinemets, and C. Li. 2018. Divergent assemblage
patterns and driving forces for bacterial and fungal communities along a glacier forefield
chronosequence. *Soil Biology and Biochemistry* **118**:207-216.

McHugh, T. A., G. W. Koch, and E. Schwartz. 2014. Minor changes in soil bacterial and fungal
community composition occur in response to monsoon precipitation in a semiarid
grassland. *Microb Ecol* **68**:370-378.

McHugh, T. A., and E. Schwartz. 2016. A watering manipulation in a semiarid grassland induced
changes in fungal but not bacterial community composition. *Pedobiologia* **59**:121-127.

Piccardi, P., B. Vessman, and S. Mitri. 2019. Toxicity drives facilitation between 4 bacterial
species. *Proc Natl Acad Sci U S A* **116**:15979-15984.

Schimel, J., T. C. Balsler, and M. Wallenstein. 2007. MICROBIAL STRESS-RESPONSE PHYSIOLOGY
AND ITS IMPLICATIONS FOR ECOSYSTEM FUNCTION. *Ecology* **88**:1386-1394.

Shade, A., H. Peter, S. D. Allison, D. L. Baho, M. Berga, H. Burgmann, D. H. Huber, S.
Langenheder, J. T. Lennon, J. B. Martiny, K. L. Matulich, T. M. Schmidt, and J.
Handelsman. 2012. Fundamentals of microbial community resistance and resilience.
*Front Microbiol* **3**:417.

Shi, S., E. E. Nuccio, Z. J. Shi, Z. He, J. Zhou, and M. K. Firestone. 2016. The interconnected
rhizosphere: High network complexity dominates rhizosphere assemblages. *Ecology*
*letters* **19**:926-936.

Sun, S., S. Li, B. N. Avera, B. D. Strahm, B. D. Badgley, and F. E. Löffler. 2017. Soil Bacterial and
Fungal Communities Show Distinct Recovery Patterns during Forest Ecosystem
Restoration. *Applied and Environmental Microbiology* **83**:e00966-00917.

Varoquaux, N., B. Cole, C. Gao, G. Pierroz, C. R. Baker, D. Patel, M. Madera, T. Jeffers, J.
Hollingsworth, J. Sievert, Y. Yoshinaga, J. A. Owiti, V. R. Singan, S. DeGraaf, L. Xu, M. J.
Blow, M. J. Harrison, A. Visel, C. Jansson, K. K. Niyogi, R. Hutmacher, D. Coleman-Derr, R.
C. O'Malley, J. W. Taylor, J. Dahlberg, J. P. Vogel, P. G. Lemaux, and E. Purdom. 2019.
Transcriptomic analysis of field-droughted sorghum from seedling to maturity reveals
biotic and metabolic responses. *Proc Natl Acad Sci U S A*:27124-27132.

Velez, P., L. Espinosa-Asuar, M. Figueroa, J. Gasca-Pineda, E. Aguirre-von-Wobeser, L. E.
Eguiarte, A. Hernandez-Monroy, and V. Souza. 2018. Nutrient Dependent Cross-
Kingdom Interactions: Fungi and Bacteria From an Oligotrophic Desert Oasis. *Front*
*Microbiol* **9**:1755.

Xu, L., D. Naylor, Z. Dong, T. Simmons, G. Pierroz, K. K. Hixson, Y. M. Kim, E. M. Zink, K. M.
Engbrecht, Y. Wang, C. Gao, S. DeGraaf, M. A. Madera, J. A. Sievert, J. Hollingsworth, D.
Birdseye, H. V. Scheller, R. Hutmacher, J. Dahlberg, C. Jansson, J. W. Taylor, P. G.
Lemaux, and D. Coleman-Derr. 2018. Drought delays development of the sorghum root
microbiome and enriches for monoderm bacteria. *Proc Natl Acad Sci U S A* **115**:E4284-
E4293.

Reviewer comments, second round –

Reviewer #1 (Remarks to the Author):

The authors have carried out a considerable revision of their manuscript, and in general, have addressed most of my concerns. I have some remaining concerns, which I detail below. I find interpreting Figure 1 and S1 difficult. I particularly struggled with the shaded vs unshaded data. Could the authors help the reader somehow, perhaps by indicating in the text discussing the figure whether they are referring to the shaded or unshaded parts of the graph?

Regarding the general concern 3 about correcting p-values with FDR. This seems an appropriate response, however, without statistics regarding how many nodes or edges were removed it is hard to assess the impact of FDR in their networks.

Regarding general concern 4: I am happy with the author's response.

Regarding general concern 5: I have some remaining concerns about calling associations/co-occurrences as interactions throughout the manuscript. The authors refer to F-F, F-B interactions etc throughout the manuscript, but this is not what they measured. The text in lines 415-425 is useful and needed, however, the authors themselves acknowledge that correlation does not equate to interaction. Ideally, they should use associations or co-occurrences instead.

Regarding general concern 6: I am happy with the author's response.

Line 214: the authors wrote "we found that the resistance to drought stress for fungal mycobiomes was consistently stronger than that for bacterial microbiomes for weeks 5 in root, weeks 4 – 6 in rhizosphere, and weeks 4 and 6 – 8 in rhizosphere". Do the authors mean ..."weeks 4 and 6 – 8 in soil"?

In some cases, the authors seem to overstate the differences between networks (to me anyway). The use of drastically/strongly enhanced co-occurrences in some cases seems inappropriate when "enhancing" alone would suffice. In the legend for figure S3: I would say that FF- co-occurrence is enhanced by drought but not necessarily drastically so. Following this, the recovery in F-F network following re-wetting seems subtle for soil (if at all) and for root. For Figure S14 legend, I find the use of "drastically" and "strongly" excessive. Likewise for enhanced in figure S15 when discussing rhizosphere F-F network.

In Fig 1 legend, the authors state "32 of 36 cases". What is each "case", I presume it is communities, and the authors should indicate that.

Line 466: It is helpful that the authors provide the total number of samples collected. However, it would be useful to know the minimum number of samples used to build a single network, and whether the number of samples used to build networks varied between the different communities, as the number of samples may affect network inference. The total number of samples collected (1026) divided by the number of communities (84 based on 48 rewetting and 36 drought) is ca. 12, which is a relatively low number of samples to build correlation networks (as indicated by Berry et al 2014, 10.3389/fmicb.2014.00219, which suggests > 25 samples per network, although I accept that papers have been published with fewer samples).

Other comments:

Line 55 (abstract): this strengthening was not always "dramatic".

Line 139: Co-occurrence network focuses on significant associations, not interactions.

Line 128: "not interaction between bacteria and fungi" (add s in interaction).

Line 185: change "form" to "from"

Lines 331-332: "Both network of AMF and other fungi and network of AMF and bacteria, when re-wetted, largely recovered". This does not seem to be the case in the rhizosphere. In (A), the rewetting panel there are fewer interactions in rhizosphere under rewetting than control, and for panel B, if there are differences they are hard to assess visually.

Lines 389-390: also could be a slower response not captured by the study.

Likewise for 405-406: could this be a temporal effect? In other words, could sampling over a longer period post rewetting show a different pattern?

Line 494: delete extra space before the full stop.

Reviewer #2 (Remarks to the Author):

All of my concerns were addressed and the revised manuscript is now exceptionally well written and clear. Further, the work represents a very important, direct, and comprehensive contribution to the field of resistance/resilience ecology as it relates to microbial communities within agricultural systems. The authors presented very detailed and attentive responses to the concerns of the reviewers and issues related to the statistical implications of the approach have also been addressed. I thoroughly enjoyed reading this revised version of the manuscript and my recommendation is for publication without further revisions.

Reviewer #3 (Remarks to the Author):

This manuscript has improved in clarity and the figures are much easier to understand. The authors have addressed most of my and the other reviewers' comments, and have done a number of additional analyses while they removed some others. However, while presenting interesting patterns, I still feel that the manuscript lacks conceptual framing and hypothesis development. Yes, it tests hypotheses that have previously been tested, but what are the new insights here? I think this lack of conceptual framing and insight is caused because the authors never, in detail, explore what these networks actually mean. Again, as I stated in my comments on the previous version, what would be interesting here is to develop hypotheses on how networks in soil, roots, and leaves would differ in their response to drought. As it stands, the manuscript reads very repetitive and does not offer a clear step forward in our understanding of network responses to drought.

However, in response to one of my other comments, it appeared that the networks in this study not only include datapoints from the 6 true field replicates, but also lump together the various time points during the progressing drought (6 time points over 6 weeks) and during the recovery period (8 timepoints over 8 weeks). This approach is not mentioned explicitly and not justified, and it seems rather inappropriate to me. It is clear that during those periods, microbial communities go through large changes (as can be seen in Fig. 1, although no information is presented on shifts in community composition here) and not only am I wondering what networks of these combined time points actually represent, as far as I am aware, no other studies constructing networks have lumped time points, which means that they can't be compared to these. This also brings me back to my most important issue, which is that it is hardly explored what these networks/ interactions actually mean ecologically.

Reviewer #4 (Remarks to the Author):

The authors have addressed all concerns and the manuscript has significantly improved. This is a great paper that will certainly catch attention in the plant-microbe community and will be cited.

**Response to reviews of our revised manuscript.**

Four reviewers responded to our revised manuscript, and all four complemented our first
revision. Two reviewers (#s 1 and 3) asked for additional revisions while the other two (#s 2 and
4) did not.

7 **FULL REVIEWER COMMENTS**

9 **Reviewer #1 (Remarks to the Author):**

The authors have carried out a considerable revision of their manuscript, and in general, have
addressed most of my concerns. I have some remaining concerns, which I detail below.

I find interpreting Figure 1 and S1 difficult. I particularly struggled with the shaded vs unshaded
data. Could the authors help the reader somehow, perhaps by indicating in the text discussing
the figure whether they are referring to the shaded or unshaded parts of the graph?

**Response:** We appreciate this comment and have revised the figure and figure legend to clarify
matters.

**Lines 791 and 795 in change-tracked manuscript: Revised legend of Figure 1 and also of**
**Figure S1:** Ecological resistance to drought stress is ... at each of the droughted weeks (weeks 3
23 – 8, the grey shaded area). Ecological resilience to rewetting is ... after rewetting were weeks 9
24 – 17 (the gold shaded area).

Regarding the general concern 3 about correcting p-values with FDR. This seems an appropriate
response, however, without statistics regarding how many nodes or edges were removed it is
hard to assess the impact of FDR in their networks.

**Response:** We assessed the impact of applying a FDR to network structure and provided the
results in the supplementary Table S3. Out of the 64 networks examined, 16 were affected by
FDR correction, and the proportion of edge removal ranged from 19.49% to 94.76% and the
proportion of vertices removal ranged from 10.84% to 90.40%. Information added in line 573 of
change-tracked manuscript

Table S3 The number and proportion of network edge and vertices removed due to FDR correction

Network	Compartment	Treatment	Period	No.edges FDR	No.edges nFDR	Edges Removed	No.vertices FDR	No.vertices nFDR	Vertices Removed
Inter-Bac-Fung	Root	Stress	Drought	10	191	94.76%	17	177	90.40%
Cross-Bac-Fung	Root	Stress	Drought	95	1130	91.59%	102	540	81.11%
Bac-Bac	Root	Stress	Drought	77	888	91.33%	79	448	82.37%
Fung-Fung	Root	Stress	Drought	8	51	84.31%	13	49	73.47%
Bac-Bac	Soil	Stress	Drought	193	848	77.24%	187	611	69.39%
Inter-Bac-Fung	Soil	Stress	Drought	52	225	76.89%	73	257	71.60%
Cross-Bac-Fung	Soil	Stress	Drought	272	1164	76.63%	263	814	67.69%
Fung-Fung	Soil	Stress	Drought	27	91	70.33%	34	86	60.47%
Inter-Bac-Fung	Soil	Control	Drought	274	408	32.84%	226	309	26.86%
Inter-Bac-Fung	Rhizosphere	Stress	Drought	161	228	29.39%	143	185	22.70%
Cross-Bac-Fung	Rhizosphere	Stress	Drought	811	1085	25.25%	439	536	18.10%
Bac-Bac	Rhizosphere	Stress	Drought	481	643	25.19%	324	395	17.97%
Cross-Bac-Fung	Soil	Control	Drought	1859	2482	25.10%	788	972	18.93%
Bac-Bac	Soil	Control	Drought	1490	1956	23.82%	636	784	18.88%
Fung-Fung	Rhizosphere	Stress	Drought	169	214	21.03%	83	95	12.63%
Fung-Fung	Soil	Control	Drought	95	118	19.49%	74	83	10.84%
Bac-Bac	Leaf	Control	Drought	43	43	0	47	47	0
Bac-Bac	Leaf	Control	Rewetting	433	433	0	79	79	0
Bac-Bac	Leaf	Stress	Drought	141	141	0	93	93	0
Bac-Bac	Leaf	Stress	Rewetting	1015	1015	0	138	138	0
Bac-Bac	Rhizosphere	Control	Drought	10234	10234	0	887	887	0
Bac-Bac	Rhizosphere	Control	Rewetting	5050	5050	0	686	686	0
Bac-Bac	Rhizosphere	Stress	Rewetting	13730	13730	0	761	761	0
Bac-Bac	Root	Control	Drought	10518	10518	0	608	608	0

Bac-Bac	Root	Control	Rewetting	2755	2755	0	348	348	0
Bac-Bac	Root	Stress	Rewetting	9030	9030	0	495	495	0
Bac-Bac	Soil	Control	Rewetting	1151	1151	0	590	590	0
Bac-Bac	Soil	Stress	Rewetting	1879	1879	0	632	632	0
Cross-Bac-Fung	Leaf	Control	Drought	122	122	0	73	73	0
Cross-Bac-Fung	Leaf	Control	Rewetting	554	554	0	117	117	0
Cross-Bac-Fung	Leaf	Stress	Drought	189	189	0	117	117	0
Cross-Bac-Fung	Leaf	Stress	Rewetting	1436	1436	0	186	186	0
Cross-Bac-Fung	Rhizosphere	Control	Drought	11116	11116	0	1036	1036	0
Cross-Bac-Fung	Rhizosphere	Control	Rewetting	7371	7371	0	896	896	0
Cross-Bac-Fung	Rhizosphere	Stress	Rewetting	16408	16408	0	894	894	0
Cross-Bac-Fung	Root	Control	Drought	12684	12684	0	714	714	0
Cross-Bac-Fung	Root	Control	Rewetting	3478	3478	0	433	433	0
Cross-Bac-Fung	Root	Stress	Rewetting	11000	11000	0	596	596	0
Cross-Bac-Fung	Soil	Control	Rewetting	1505	1505	0	760	760	0
Cross-Bac-Fung	Soil	Stress	Rewetting	2127	2127	0	749	749	0
Inter-Bac-Fung	Leaf	Control	Drought	3	3	0	5	5	0
Inter-Bac-Fung	Leaf	Control	Rewetting	82	82	0	46	46	0
Inter-Bac-Fung	Leaf	Stress	Drought	2	2	0	4	4	0
Inter-Bac-Fung	Leaf	Stress	Rewetting	331	331	0	96	96	0
Inter-Bac-Fung	Rhizosphere	Control	Drought	777	777	0	391	391	0
Inter-Bac-Fung	Rhizosphere	Control	Rewetting	1529	1529	0	437	437	0
Inter-Bac-Fung	Rhizosphere	Stress	Rewetting	2398	2398	0	474	474	0
Inter-Bac-Fung	Root	Control	Drought	1840	1840	0	417	417	0
Inter-Bac-Fung	Root	Control	Rewetting	619	619	0	246	246	0
Inter-Bac-Fung	Root	Stress	Rewetting	1836	1836	0	409	409	0
Inter-Bac-Fung	Soil	Control	Rewetting	161	161	0	145	145	0
Inter-Bac-Fung	Soil	Stress	Rewetting	167	167	0	157	157	0

Fung-Fung	Leaf	Control	Drought	76	76	0	24	24	0
Fung-Fung	Leaf	Control	Rewetting	39	39	0	31	31	0
Fung-Fung	Leaf	Stress	Drought	46	46	0	22	22	0
Fung-Fung	Leaf	Stress	Rewetting	90	90	0	42	42	0
Fung-Fung	Rhizosphere	Control	Drought	105	105	0	77	77	0
Fung-Fung	Rhizosphere	Control	Rewetting	792	792	0	159	159	0
Fung-Fung	Rhizosphere	Stress	Rewetting	280	280	0	94	94	0
Fung-Fung	Root	Control	Drought	326	326	0	91	91	0
Fung-Fung	Root	Control	Rewetting	104	104	0	64	64	0
Fung-Fung	Root	Stress	Rewetting	134	134	0	69	69	0
Fung-Fung	Soil	Control	Rewetting	193	193	0	131	131	0
Fung-Fung	Soil	Stress	Rewetting	81	81	0	75	75	0

Regarding general concern 4: I am happy with the author's response.

**Response: Thank you!**

Regarding general concern 5: I have some remaining concerns about calling associations/co-
occurrences as interactions throughout the manuscript. The authors refer to F-F, F-B
interactions etc throughout the manuscript, but this is not what they measured. The text in
lines 415-425 is useful and needed, however, the authors themselves acknowledge that
correlation does not equate to interaction. Ideally, they should use associations or co-
occurrences instead.

**Response: We agree with the reviewer and now use association instead of interaction**
**throughout the revised manuscript. Revised in lines 47, 114, 115, 121, 130, 133, 134, 142, 153,**
**180, 248-252, 276, 311, 363, 383, 386, 389, 424, 433, 434, 451-456, 467, 495, 579, 813, 833,**
**834 in change-tracked manuscript.**

Regarding general concern 6: I am happy with the author's response.

**Response: Thank you!**

Line 214: the authors wrote "we found that the resistance to drought stress for fungal
mycobiomes was consistently stronger than that for bacterial microbiomes for weeks 5 in root,
56 weeks 4 – 6 in rhizosphere, and weeks 4 and 6 – 8 in rhizosphere". Do the authors
mean ..."weeks 4 and 6 – 8 in soil"?

**Response: We are grateful that the reviewer caught our error. We corrected it in the revised**
**manuscript in line 230 of change-tracked manuscript.**

In some cases, the authors seem to overstate the differences between networks (to me
anyway). The use of drastically/strongly enhanced co-occurrences in some cases seems
inappropriate when "enhancing" alone would suffice. In the legend for figure S3: I would say
that FF- co-occurrence is enhanced by drought but not necessarily drastically so. Following this,
the recovery in F-F network following re-wetting seems subtle for soil (if at all) and for root. For
Figure S14 legend, I find the use of "drastically" and "strongly" excessive. Likewise for enhanced
in figure S15 when discussing rhizosphere F-F network.

**Response: We agree with the reviewer and now, to avoid overstating our results, we have**
**removed the words 'drastically', or 'strongly' in the legend of Figure S3, S14 and S15, and in**
**lines 55, 267, 432 and 832 of the change-tracked manuscript.**

In Fig 1 legend, the authors state "32 of 36 cases". What is each "case", I presume it is
communities, and the authors should indicate that.

**Response: We appreciate the reviewer finding this ambiguity and we have changed 'cases' into**
**'communities' in the legend of Fig 1 in line 797 of change-tracked manuscript.**

Line 466: It is helpful that the authors provide the total number of samples collected. However,
it would be useful to know the minimum number of samples used to build a single network, and
whether the number of samples used to build networks varied between the different

communities, as the number of samples may affect network inference. The total number of
samples collected (1026) divided by the number of communities (84 based on 48 rewetting and
36 drought) is ca. 12, which is a relatively low number of samples to build correlation networks
(as indicated by Berry et al 2014, 10.3389/fmicb.2014.00219, which suggests > 25 samples per
network, although I accept that papers have been published with fewer samples).

**Response:** We agree with the reviewer and we, too, were concerned about the relatively low
number of plots (six plots for each of the three treatments) in our study. Therefore, we
analyzed networks for each period and treatment separately. Thus, the drought period network
was based on 36 communities (6 plots * 6 time points) and the rewetting period network was
based on 48 communities (6 plots * 8 time points). We now provide this information in lines
567-568 of change-tracked manuscript

**Revised:** We analyzed networks for each period and treatment separately, following previous
studies⁶²⁻⁶⁵, to assure > 25 communities per network⁶⁶. Thus, the drought-period network was
based on 36 communities (6 plots * 6 time points) and the rewetting period network was based
on 48 communities (6 plots * 8 time points).

Other comments:

Line 55 (abstract): this strengthening was not always “dramatic”.

**Response:** We agree with the reviewer and have removed ‘dramatically’ in line 55 of change-
tracked manuscript.

Line 139: Co-occurrence network focuses on significant associations, not interactions.

**Response:** We agree with the reviewer and have changed ‘interactions’ into ‘associations’ lines
104 47, 114, 115, 121, 130, 133, 134, 142, 153, 180, 248-252, 276, 311, 363, 383, 386, 389, 424, 433,
434, 451-456, 467, 495, 579, 813, 833, 834 in change-tracked manuscript.

Line 128: “not interaction between bacteria and fungi” (add s in interaction).

**Response:** We appreciate the reviewer catching our error in English usage. Note the word
interaction has been changed into association according to your above comment. We added an
‘s’ to association in line 130 of change-tracked manuscript .

Line 185: change “form” to “from”

**Response:** We now see that our use of form was ambiguous. We have changed ‘growth form’
to ‘form of growth’ in line 200 of change-tracked manuscript.

Lines 331-332: “Both network of AMF and other fungi and network of AMF and bacteria, when
re-wetted, largely recovered”. This does not seem to be the case in the rhizosphere. In (A), the
rewetting panel there are fewer interactions in rhizosphere under rewetting than control, and
for panel B, if there are differences they are hard to assess visually.

**Response:** We appreciate this comment from the reviewer and now more accurately describe
the results in lines 354-358 of change-tracked manuscript.

**Revised:** Networks in roots and soil of both AMF and other fungi and AMF and bacteria, when
re-wetted, largely recovered their pre-drought complexity. In rhizosphere, however, the
network of AMF and other fungi and was less complex in rewetting than the control (Fig. 5A),
and the network of AMF and bacteria, when re-wetted, largely recovered was not different
from the control (Fig. 55B).

Lines 389-390: also could be a slower response not captured by the study.

**Response:** We agree with the reviewer and have added this explanation in lines 427-429 of
change-tracked manuscript.

**Added text:** These results could also be explained by a slower response in rhizosphere or soil
that was not captured over the period of our study.

Likewise for 405-406: could this be a temporal effect? In other words, could sampling over a
longer period post rewetting show a different pattern?

**Response:** We agree with the reviewer and have added this information here in line 445-447 of
change-tracked manuscript.

**Added:** Also, it's unclear whether a different pattern would be observed if the micro- and
mycobiomes were investigated over longer periods.

Line 494: delete extra space before the full stop.

**Response:** We thank the reviewer for catching this typo and we have removed the extra space
in line 542 of change-tracked manuscript.

**Reviewer #2 (Remarks to the Author):**

All of my concerns were addressed and the revised manuscript is now exceptionally well
written and clear. Further, the works represents an very important, direct, and comprehensive
contribution to the field of resistance/resilience ecology as it relates to microbial communities
within agricultural systems. The authors presented very detailed and attentive responses to the
concerns of the reviewers and issues related to the statistical implications of the approach have
also been addressed. I thoroughly enjoyed reading this revised version of the manuscript and
my recommendation is for publication without further revisions.

**Response:** we are happy to learn that the reviewer is satisfied with our efforts in revision.

**Reviewer #3 (Remarks to the Author):**

This manuscript has improved in clarity and the figures are much easier to understand. The
authors have addressed most of my and the other reviewers' comments, and have done a
number of additional analyses while they removed some others. However, while presenting

interesting patterns, I still feel that the manuscript lacks conceptual framing and hypothesis
development. Yes, it tests hypotheses that have previously been tested, but what are the new
insights here? I think this lack of conceptual framing and insight is caused because the authors
never, in detail, explore what these networks actually mean. Again, as I stated in my comments
on the previous version, what would be interesting here is to develop hypotheses on how
networks in soil, roots, and leaves would differ in their response to drought. As is stands, the
manuscript reads very repetitive and does not offer a clear step forward in our understanding
of network responses to drought.

**Response:** We welcome the opportunity to add more text about the ecological interpretation
of our results. Before presenting new text, we want to point out that we framed three
hypotheses and tested them with one traditional and two new approaches and enough data to
fairly establish significance. Although we are averse to speculation, we did include some text
that considered biological phenomena responsible for our results. Here are seven examples:

L166 Identification of key network elements, in this case modules or hubs, may facilitate
practical application of microbial networks to modern agriculture. Modules, the highly inter-
connected sub-structures within networks, may represent ecological units comprising highly
interacting members (Newman 2006). Network hubs, microbes located in the central position
of the network, and modular hubs, microbes located in the central position within a module, or
connectors, which link different modules, are both disproportionately important in structuring
microbial communities (Agler et al. 2016). Artificial inoculation of these hub taxa might provide
a means of directing the microbial community, or key modules within the community, to
reduce inputs or improve yields for modern agriculture (Toju et al. 2018).

[revised manuscript text omitted]

**Here are added sections that provide insights (speculation?) about our results and the**
**underlying biology.**

Added Introduction in lines 157-165 of change-tracked manuscript: Should we expect that the
microbiomes and mycobiomes that inhabit the different plant compartments (leaf, root,
rhizosphere, and soil) will respond similarly to drought? Existing literature does not answer this
question because previous investigations of co-occurrence networks are largely limited in one
compartment (Table S1). By considering all four compartments in previous reports, we showed
that drought responses of fungal and bacterial communities are most pronounced in root,
followed by rhizosphere and, lastly, soil and leaves, where the responses were much weaker
266 ^{10,11} (Fig. 1). Guided by these results, here we extend the network hypothesis to all four plant
compartments: drought disrupts microbial network more strongly in root than rhizosphere, soil
and leaf compartments.

Added Discussion in lines 395-397 and 404-406 of change-tracked manuscript: At the dimension
of plant compartments, drought disrupted root networks more strongly than those of other
compartments... This result may reflect stronger reduction of plant resources in the root, which
would lead to stronger disruptions of bacterial and fungal networks in this compartment.

We also added discussion on the step forward in our understanding of network responses to
drought in lines 407-411 of change-tracked manuscript: Previous studies also report disruption
by drought of soil bacterial co-occurrence networks along natural arid gradients ^{41,42}, but
another study did not report any effect of drought on soil fungal co-occurrence networks in
potted plants ¹⁹. Our study of field-grown plants shows that drought can enhance as well as
disrupt microbiome networks, emphasizing the positive role that bacterial and fungal
communities can play in plant drought response.

We also added discussion on the ecological meaning of networks in lines 457-459 of change-
tracked manuscript: The detected associations in networks may composed of a mixture of real
and false interactions, of direct and indirect interactions, and of physical and chemical
interactions.....While the exact nature of correlative associations cannot be recognized by our
amplicon-based method, the changes in network complexity and detections of network hubs
can be used to infer ecological function.

However, in response to one of my other comments, it appeared that the networks in this study
not only include datapoints from the 6 true field replicates, but also lump together the various
time points during the progressing drought (6 time points over 6 weeks) and during the
recovery period (8 timepoints over 8 weeks). This approach is not mentioned explicitly and not
justified, and it seems rather inappropriate to me. It is clear that during those periods, microbial
communities go through large changes (as can be seen in Fig. 1, although no information is
presented on shifts in community composition here) and not only am I wondering what

networks of these combined time points actually represent, as far as I am aware, no other
studies constructing networks have lumped time points, which means that they can't be
compared to these.

**Response about explicit text on analysis of more than one time point:** We appreciate the
reviewer's concern and have added text show how the analyses were conducted, as shown
above and restated below.

**Revised** in lines 567-568 of change-tracked manuscript: We analyzed networks for each period
and treatment separately, following previous studies⁶²⁻⁶⁵, to assure > 25 communities per
network⁶⁶. Thus, the drought-period network was based on 36 communities (6 plots * 6 time
points) and the rewetting period network was based on 48 communities (6 plots * 8 time
points).

**Response about the practice of analyzing sequential time points:** To alleviate the reviewer's
concern ("... as far as I am aware, no other studies constructing networks have lumped time
points."), our search on google scholar returned numerous studies that construct network using
samples of different time points as shown by these examples (full references at the end of this
document):

36 time points (Lejal et al. 2021)
35 / 120 time points (Fuhrman et al. 2015)
253 / 365 time points (Faust et al. 2018)
72 time points (Gilbert et al. 2012)
15 time points (Pinto et al. 2014)
3 time points (Dunphy et al. 2019)
5 time points (Shade et al. 2013)
4 time points (Liu and Howell 2021)
3 time points (Jiao et al. 2017)
5 time points (Carini et al. 2020)

This also bring me back to my most important issue, which is that it is hardly explored what
these networks/ interactions actually mean ecologically.

**Response given just above.**

**Reviewer #4 (Remarks to the Author):**

The authors have addressed all concerns and the manuscript has significantly improved. This is
a great paper that will certainly catch attention in the plant-microbe community and will be
cited.

**Response:** we are happy to learn that the reviewer is satisfied with our efforts in revision.

References

- Carini, P., M. Delgado-Baquerizo, E.-L. S. Hinckley, H. Holland-Moritz, T. E. Brewer, G. Rue, C. Vanderburgh, D. McKnight, N. Fierer, and J. K. Jansson. 2020. Effects of Spatial Variability and Relic DNA Removal on the Detection of Temporal Dynamics in Soil Microbial Communities. *mBio* **11**:e02776-02719.
- Dunphy, C. M., T. C. Gouhier, N. D. Chu, and S. V. Vollmer. 2019. Structure and stability of the coral microbiome in space and time. *Scientific Reports* **9**:6785.
- Faust, K., F. Bauchinger, B. Laroche, S. de Buyl, L. Lahti, A. D. Washburne, D. Gonze, and S. Widder. 2018. Signatures of ecological processes in microbial community time series. *Microbiome* **6**:120.
- Fuhrman, J. A., J. A. Cram, and D. M. Needham. 2015. Marine microbial community dynamics and their ecological interpretation. *Nature Reviews Microbiology* **13**:133-146.
- Gilbert, J. A., J. A. Steele, J. G. Caporaso, L. Steinbrück, J. Reeder, B. Temperton, S. Huse, A. C. McHardy, R. Knight, I. Joint, P. Somerfield, J. A. Fuhrman, and D. Field. 2012. Defining seasonal marine microbial community dynamics. *The ISME Journal* **6**:298-308.
- Jiao, S., Z. Zhang, F. Yang, Y. Lin, W. Chen, and G. Wei. 2017. Temporal dynamics of microbial communities in microcosms in response to pollutants. *Molecular Ecology* **26**:923-936.
- Lejal, E., J. Chiquet, J. Aubert, S. Robin, A. Estrada-Peña, O. Rue, C. Midoux, M. Mariadassou, X. Bailly, A. Cougoul, P. Gasqui, J. F. Cosson, K. Chalvet-Monfray, M. Vayssier-Taussat, and T. Pollet. 2021. Temporal patterns in *Ixodes ricinus* microbial communities: an insight into tick-borne microbe interactions. *Microbiome* **9**:153.
- Liu, D., and K. Howell. 2021. Community succession of the grapevine fungal microbiome in the annual growth cycle. *Environmental Microbiology* **23**:1842-1857.
- Pinto, A. J., J. Schroeder, M. Lunn, W. Sloan, L. Raskin, and M. A. Moran. 2014. Spatial-Temporal Survey and Occupancy-Abundance Modeling To Predict Bacterial Community Dynamics in the Drinking Water Microbiome. *mBio* **5**:e01135-01114.
- Shade, A., P. S. McManus, and J. Handelsman. 2013. Unexpected diversity during community succession in the apple flower microbiome. *mBio* **4**:e00602-00612.

Reviewer comments, third round –

Reviewer #1 (Remarks to the Author):

I am happy with the changes to the manuscript. The methodological questions I raised were clarified, and some of the language and writing changed to reflect my comments. I enjoyed reading the manuscript and appreciate that the authors were able to accommodate my concerns.

I have read the response from the author. The response was basically to say that the LSA method (which was developed to deal with time series data but has some problems as the authors pointed out) is not appropriate for their data. Also, they mention that a previous study published in Nature (Dai et al) used the same method (Spearman rank correlation) for a dataset that was also collected along a time series. This is a valid response.

However, the authors did not specifically respond how their method may deal with specific issues that time series data cause for network analysis. In particular, time series data can lead to considerable issues with temporal autocorrelation (for more details: <https://www.frontiersin.org/articles/10.3389/fgene.2020.00310/full>). More likely, their data includes some temporal autocorrelation, but spurious correlations are a limitation of network analysis which does not necessarily compromise the work if the author took the steps to minimise the artefacts. The author's carried out false rate discovery correction of their data, which removes weaker/spurious correlations (although, strangely, in many of their networks FDR correction did not remove any correlations or nodes from their network).

The authors cited the work by Dai et al (2022) to demonstrate how their network methodology is valid. However, this study (Dai et al), while using spearman rank correlations, removed weaker correlations using the Random Matrix Theory approach, and they compared their networks to random networks generated from their data, which gives a measure of how robust their networks are.

In short, it is hard for me to assess how much temporal autocorrelation may have affected their networks, it may not be a problem, but the authors did not explain how they took temporal autocorrelation into account. They chose FDR to remove weaker correlations, which is a valid approach, but this correction did not affect many of their networks. Unless I am missing something, comparing the properties of their true networks with those of random networks generated from their data would give more confidence in the robustness of their networks.

REVIEWER COMMENTS

Reviewer #1 (Remarks to the Author):

I am happy with the changes to the manuscript. The methodological questions I raised were clarified, and some of the language and writing changed to reflect my comments. I enjoyed reading the manuscript and appreciate that the authors were able to accommodate my concerns.

I have read the response from the author. The response was basically to say that the LSA method (which was developed to deal with time series data but has some problems as the authors pointed out) is not appropriate for their data. Also, they mention that a previous study published in Nature (Dai et al) used the same method (Spearman rank correlation) for a dataset that was also collected along a time series. This is a valid response.

However, the authors did not specifically respond how their method may deal with specific issues that time series data cause for network analysis. In particular, time series data can lead to considerable issues with temporal autocorrelation (for more details: <https://www.frontiersin.org/articles/10.3389/fgene.2020.00310/full>). More likely, their data includes some temporal autocorrelation, but spurious correlations are a limitation of network analysis which does not necessarily compromise the work if the author took the steps to minimise the artefacts. The author's carried out false rate discovery correction of their data, which removes weaker/spurious correlations (although, strangely, in many of their networks FDR correction did not remove any correlations or nodes from their network).

The authors cited the work by Dai et al (2022) to demonstrate how their network methodology is valid. However, this study (Dai et al), while using spearman rank correlations, removed weaker correlations using the Random Matrix Theory approach, and they compared their networks to random networks generated from their data, which gives a measure of how robust their networks are.

In short, it is hard for me to assess how much temporal autocorrelation may have affected their networks, it may not be a problem, but the authors did not explain how they took temporal autocorrelation into account. They chose FDR to remove weaker correlations, which is a valid approach, but this correction did not affect many of their networks. Unless I am missing something, comparing the properties of their true networks with those of random networks generated from their data would give more confidence in the robustness of their networks.

Response: We have address the reviewer's concern about undetected autocorrelation using approaches suggested by the reviewer, one in Coenen et al 2020 (<https://www.frontiersin.org/articles/10.3389/fgene.2020.00310/full>) and another in Dai et al. (2022). As you can see, below, these additional analyses of our data do not reveal significant autocorrelation and do not affect our findings.

We will begin with the approach described in Figure 1 of Coenen et al 2020, who wrote: “In Figure 1, we show how autocorrelation leads to high incidences of spurious correlations among independent time-series (with 100 random walks).”

To determine if our dataset suffered from similar temporal autocorrelation, we used the approach of Coenen et al 2020 to search for spurious associations for **6 random walks** (mimicking the drought period) and **8 random walks** (mimicking the rewetting period) of 6 time series (mimicking our six replicating samples). The results from 10 runs for each test show that our datasets are not influenced significantly by temporal autocorrelation. For the drought period, at most 0-1 of 15 correlations showed significant association (Fig. S14A-B) and for rewetting at most 1-3 of 15 correlations showed significant association (Fig. S14C-D).

We propose adding text to the manuscript and a supplemental figure S14 with the results from one run, as shown below.

Added Text in lines 557-563: Concern about temporal autocorrelation, leading to spurious correlations among independent time-series, led us to use the approach of Coenen, et al. ⁶⁷ to simulate 6 random walks (mimicking the drought period) and 8 random walks (mimicking the rewetting period) of 6 time series (mimicking our six replicating samples). We were unable to detect significant temporal autocorrelation among the 15 comparisons of six, random time series for either the drought period (≤ 1 significant association, Fig. S14A-B) or rewetting ($\leq 1-3$ significant associations, Fig. S14C-D).

Fig. S14 Minimal spurious association was detected in using the approach of Coenen et al 2020 from 6 and 8 independent random walks over 6 temporal series. The analysis was repeated 10 times and results from one run are shown here. (A) Six time-series of six independent random walks mimicking the drought period. (B) For the 15 correlations among six time series of six independent random walks, at most 0-1 significant spurious associations were detected (none were found in this example). (C) Six time-series of eight independent random walks mimicking the rewetting period. (D) For the 15 correlations among six time series of eight independent random walks, at most 1-3 significant spurious associations were detected (The one in this run is marked with an asterisk in the example).

From the Dai et al. (2022) publication suggested by the reviewer, we added new analyses of network construction using the Random Matrix Theory (RMT) approach and random network comparison as implemented in the Molecular Ecological Network Analyses Pipeline (MENAP).

First, as with Dai et al 2022, we used the MENAP to comparison the empirical network against random networks, finding that all networks are non-random (Table S4).

We propose adding text to our manuscript and a supplemental table S4, as shown below.

Added text in lines 568-570: In addition to FDR, we used Random Matrix Theory (RMT) to assess the robustness of correlations as implemented in the Molecular Ecological Network Analyses Pipeline (MENAP)⁷⁰. We found that all empirical networks were non-random (Table S4).

Table S4 Non-random topological features indicated by comparing empirical network against random networks

Compartment	Treatment	Period	Network Indexes	Observation	Random network (mean \pm sd)*	P value
Root	Control	Drought	Average clustering coefficient	0.404	0.221 \pm 0.012	1.48E-124
Root	Control	Rewetting	Average clustering coefficient	0.427	0.25 \pm 0.013	1.08E-119
Root	Stress	Drought	Average clustering coefficient	0.094	0.016 \pm 0.005	1.57E-125
Root	Stress	Rewetting	Average clustering coefficient	0.461	0.278 \pm 0.012	1.48E-124
Root	Control	Drought	Average path distance	2.916	2.666 \pm 0.024	3.21E-108
Root	Control	Rewetting	Average path distance	3.077	2.656 \pm 0.027	1.64E-125
Root	Stress	Drought	Average path distance	6.065	4.533 \pm 0.08	2.46E-134
Root	Stress	Rewetting	Average path distance	3.111	2.554 \pm 0.021	2.47E-148
Root	Control	Drought	Transitivity	0.424	0.253 \pm 0.006	2.02E-151
Root	Control	Rewetting	Transitivity	0.387	0.256 \pm 0.007	2.40E-133
Root	Stress	Drought	Transitivity	0.29	0.03 \pm 0.006	1.94E-169
Root	Stress	Rewetting	Transitivity	0.445	0.296 \pm 0.006	1.68E-145
Rhizosphere	Control	Drought	Average clustering coefficient	0.267	0.074 \pm 0.01	1.14E-134
Rhizosphere	Control	Rewetting	Average clustering coefficient	0.321	0.141 \pm 0.011	1.39E-127
Rhizosphere	Stress	Drought	Average clustering coefficient	0.162	0.046 \pm 0.007	4.00E-128
Rhizosphere	Stress	Rewetting	Average clustering coefficient	0.45	0.3 \pm 0.013	1.34E-112
Rhizosphere	Control	Drought	Average path distance	4.449	3.261 \pm 0.05	1.33E-143
Rhizosphere	Control	Rewetting	Average path distance	3.804	2.99 \pm 0.034	6.26E-144
Rhizosphere	Stress	Drought	Average path distance	4.642	3.587 \pm 0.042	5.40E-146
Rhizosphere	Stress	Rewetting	Average path distance	2.921	2.62 \pm 0.022	6.69E-120
Rhizosphere	Control	Drought	Transitivity	0.283	0.098 \pm 0.008	1.94E-142
Rhizosphere	Control	Rewetting	Transitivity	0.354	0.175 \pm 0.008	5.06E-141
Rhizosphere	Stress	Drought	Transitivity	0.373	0.089 \pm 0.007	1.32E-166
Rhizosphere	Stress	Rewetting	Transitivity	0.365	0.266 \pm 0.006	6.14E-128
Soil	Control	Drought	Average clustering coefficient	0.175	0.041 \pm 0.009	1.58E-123

Soil	Stress	Drought	Average clustering coefficient	0.135	0.011 ± 0.004	4.89E-155
Soil	Stress	Rewetting	Average clustering coefficient	0.166	0.017 ± 0.004	6.21E-163
Soil	Control	Rewetting	Average clustering coefficient	0.169	0.014 ± 0.004	1.25E-164
Soil	Control	Drought	Average path distance	4.586	3.664 ± 0.057	4.37E-127
Soil	Stress	Drought	Average path distance	6.039	4.72 ± 0.099	9.21E-119
Soil	Stress	Rewetting	Average path distance	5.734	4.254 ± 0.054	9.66E-150
Soil	Control	Rewetting	Average path distance	6.379	4.48 ± 0.049	1.23E-164
Soil	Control	Drought	Transitivity	0.374	0.067 ± 0.008	3.26E-164
Soil	Stress	Drought	Transitivity	0.249	0.022 ± 0.006	1.33E-163
Soil	Stress	Rewetting	Transitivity	0.266	0.028 ± 0.005	1.78E-173
Soil	Control	Rewetting	Transitivity	0.268	0.023 ± 0.004	2.56E-184
Leaf	Control	Rewetting	Average clustering coefficient	0.4	0.383 ± 0.018	1.18E-17
Leaf	Stress	Rewetting	Average clustering coefficient	0.377	0.338 ± 0.018	1.69E-43
Leaf	Control	Rewetting	Average path distance	4.093	2.492 ± 0.048	3.49E-158
Leaf	Stress	Rewetting	Average path distance	3.087	2.584 ± 0.034	2.97E-123
Leaf	Control	Rewetting	Transitivity	0.644	0.483 ± 0.012	4.62E-119
Leaf	Stress	Rewetting	Transitivity	0.536	0.399 ± 0.01	5.87E-120

Random networks were generated at the Molecular Ecological Network Analyses Pipeline (MENAP) by randomly rewiring all the links while keeping the numbers of nodes and links of the empirical network.

Next, we compared the association of networks based on Spearman correlations as filtered by either the FDR or RMT approaches. As shown in the following figure of average degree, the results of these two different methods are consistent. The results of the two methods continue to support our first conclusion, that drought in general disrupts microbial networks. This result was found in 11 of 13 FDR networks, and 10 of 13 RMT networks. There was only one inconsistent case, concerning roots during drought, where the FF network showed disruption using the FDR approach but was unchanged using the RMT approach. We propose adding text to our manuscript a supplemental figure S15, as shown below.

Added text in lines 571-573: We then compared the association networks based on Spearman correlations as filtered by either the FDR or RMT approaches, finding that results of these two different methods are consistent in terms of drought response (Fig. S15-S16).

Fig. S15 Consistent responses to drought of average degree of association networks based on Spearman correlations as filtered by either the false discovery rate (FDR) or random matrix theory (RMT) approach. Note that in only one case, roots, is there disagreement where the FF network showed disruption using the FDR approach but was unchanged using the RMT approach.

Finally, neither did application of the new, RMT analyses affect our second conclusion, that co-occurrence networks among functional guilds of rhizosphere fungi and leaf bacteria were dramatically strengthened by drought, because these same strengthening is found with both approaches. We propose adding a supplemental figure S16, as shown below.

Fig. S16 Spearman Rho co-occurrence networks of rhizosphere fungi and leaf bacteria were dramatically strengthened by drought, whether measured by FDR- or RMT-based approach.

To reiterate, both FDR and RMT approaches support the key findings that: (i) In general, drought disrupts microbial networks based on significant positive correlations among bacteria, among fungi and between bacteria and fungi. (ii) In contrast, co-occurrence networks among functional guilds of rhizosphere fungi and leaf bacteria were dramatically strengthened by drought.